evolution/biomechanics

cadaver, tibio-calcaneal coupling, bipedal locomotion, foot kinematics

**Author for correspondence:**
Naomichi Ogihara
e-mail: ogihara@bs.s.u-tokyo.ac.jp

†These authors contributed equally to this study.

# Comparative radiographic analysis of three-dimensional innate mobility of the foot bones under axial loading of humans and African great apes

Takuo Negishi[1,†], Kohta Ito[2,†], Koh Hosoda[3], Takeo Nagura[4], Tomohiko Ota[4], Nobuaki Imanishi[4], Masahiro Jinzaki[4], Motoharu Oishi[5] and Naomichi Ogihara[1]

[1]Department of Biological Science, Graduate School of Science, The University of Tokyo, 7-3-1, Hongo, Bunkyo-ku, Tokyo 113-0033, Japan
[2]Graduate School of Human Sciences, Osaka University, Suita, Osaka Japan
[3]Graduate School of Engineering Science, Osaka University, Toyonaka, Japan
[4]School of Medicine, Keio University, Tokyo, Japan
[5]School of Veterinary Medicine, Azabu University, Sagamihara, Japan

NO, 0000-0002-1697-9263

The human foot is considered to be morphologically adapted for habitual bipedal locomotion. However, how the mobility and mechanical interaction of the human foot with the ground under a weight-bearing condition differ from those of African great apes is not well understood. We compared three-dimensional (3D) bone kinematics of cadaver feet under axial loading of humans and African great apes using a biplanar X-ray fluoroscopy system. The calcaneus was everted and the talus and tibia were internally rotated in the human foot, but such coupling motion was much smaller in the feet of African great apes, possibly due to the difference in morphology of the foot bones and articular surfaces. This study also found that the changes in the length of the longitudinal arch were larger in the human foot than in the feet of chimpanzees and gorillas, indicating that the human foot is more deformable, possibly to allow storage and release of the elastic energy during locomotion. The coupling motion of the calcaneus and the tibia, and the larger capacity

to be flattened due to axial loading observed in the human foot are possibly morphological adaptations for habitual bipedal locomotion that has evolved in the human lineage.

# 1. Introduction

The anatomy of the human foot complex is quite different from that of the biologically closest living species, African great apes, owing to morphological adaptations for obligate bipedal walking that have evolved in the human lineage [1–4]. For example, loss of opposable halluces [1,5], the presence of a longitudinal arch [6,7], formation of a large, robust calcaneus [8,9], increased rigidity of the midfoot [10–12] and shortening of the phalanges [13] are some of the morphological features that distinguish the human foot from that of the African great apes that are considered adaptive for generation of efficient and stable bipedal locomotion. The foot of the African great apes, on the other hand, is essentially a grasping organ as in other primates, retaining features generally regarded as adaptations for locomotion in arboreal environments requiring grasping capabilities [14–16]. Therefore, locomotor behaviour of fossil hominins has been generally reconstructed based on assessment of morphological affinities of the fossil bones with those of humans or African great apes (see DeSilva *et al.* [17] for review). However, quantitative analysis of form–function relationships of the foot bones is still challenging. How the specialized morphological features of the human foot facilitate generation of stable and energetically efficient bipedal locomotion has not been fully elucidated due to the complexity of the foot skeletal system, as well as the difficulty associated with the measurement of the foot bones during locomotion, which are invisible due to soft tissues surrounding the foot bones.

To develop a foundational understanding of the form–function relationship of the foot bones, we analysed the three-dimensional (3D) kinematics of the human foot under an axial loading condition using cadaver specimens in a previous study [18]. Using a biplane X-ray fluoroscopy system, we characterized the innate mobility of the human foot determined by its morphology and structure, and we found that axial loading of the human foot resulted in eversion of the calcaneus and internal rotation of the talus and tibia due to the innate morphology of the human foot [18]. This kinematic coupling of the calcaneus and tibia, the so-called tibio-calcaneal coupling, has been documented previously [19–21], but our study clarified the detailed mechanism underlying the coupling motion, and we proposed a hypothesis that the structurally embedded tibio-calcaneal coupling motion could be one of the derived features of the human foot that might facilitate the generation of efficient and robust bipedal locomotion. During the stance phase of human gait, the ground reaction moment around the vertical axis of the ground is generated in the direction of internal rotation in walking [22–26] and running [27,28]. The structurally embedded internal rotation of the tibia generated during weight-bearing could induce vertical moment, possibly advantageous to balance yaw moment acting on the body during human bipedal locomotion due to trunk rotation and leg swing. However, no studies have previously investigated whether such morphologically embedded tibio-calcaneal coupling observed in the human foot is also present in the feet of African great apes or whether it is a human-unique trait.

In the present study, we introduce new data on foot bone mobility during axial loading in chimpanzees and gorillas based on cadaver specimens using a biplane X-ray fluoroscopy system and a model registration technique, comparing these findings to those of our previous study on humans [18]. Our aim was to understand how differences in foot morphology affect the way it mechanically interacts with the ground during axial loading and its implications on the evolution of bipedal locomotion. Specifically, cadaver feet of humans, chimpanzees and gorillas were axially loaded up to 588 N by putting weight on the shaft-connected foot, and the 3D movements of the foot bones and tibia were quantified using biplane fluoroscopy and a model registration method (figure 1). We aligned the tibia vertically for all three species because during quiet standing, the tibia is oriented approximately vertically in chimpanzees as in humans [29], even though African apes walk with bent hips and bent knees. Furthermore, if the tibia is tilted in the sagittal plane and the vertical force is applied to the tibia, moment around the ankle joint is generated, which should be counterbalanced by applying additional force to the tibia to maintain static balance of force and moment. This makes the loading condition very different from that of the vertical tibia, making interspecific comparisons impossible. Therefore, the tibia must be set vertical for all three species. This new information on how foot bone morphology affects the mobility and mechanical interaction of the foot in African great apes will add to the findings of our previous study on humans to help reconstruct locomotor behaviour of fossil hominins. Although detailed foot kinematics during locomotion in chimpanzees [30,31], as well

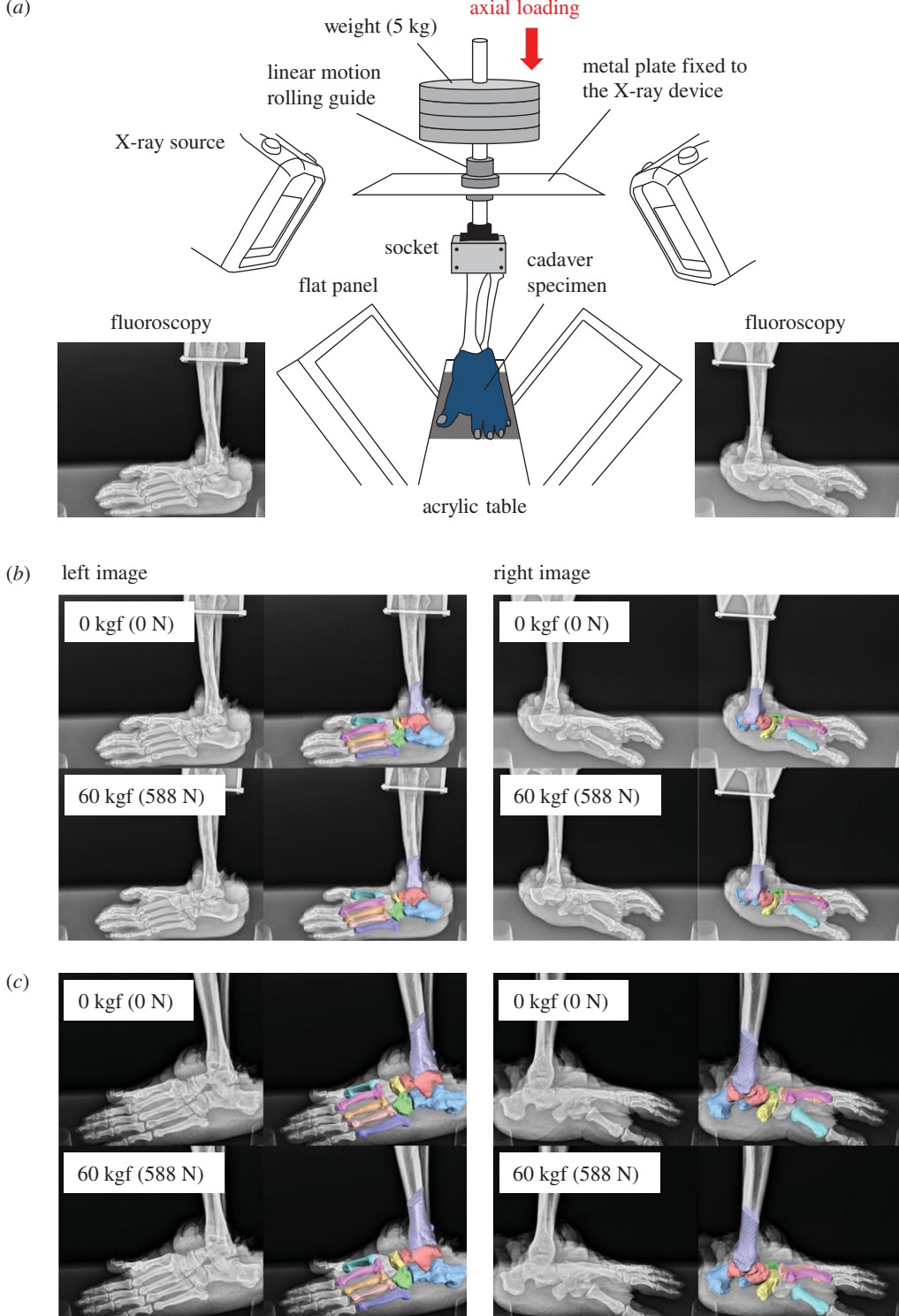

**Figure 1.** Experimental set-up to measure 3D kinematics of the foot bones under axial loading using biplane fluoroscopy (*a*) and automatic model registration of 3D surface models of the bones to biplane fluoroscopic images in one representative chimpanzee (*b*) and gorilla (*c*). The biplane fluoroscopic system consists of two X-ray sources and corresponding detector panels positioned in a quasi-orthogonal arrangement. The specimen was fixed to the shaft using a 3D-printed socket (mould). The proximal end of the lower limb was sandwiched by the anterior and posterior moulds and tightly screwed to an aluminium holder connected in line with the shaft.

as passive midfoot joint mobility of primate feet [32,33], have been reported recently, no previous studies have attempted to directly capture the 3D movements of the chimpanzee and gorilla foot bones during axial loading to clarify the innate mobility of the foot in these species.

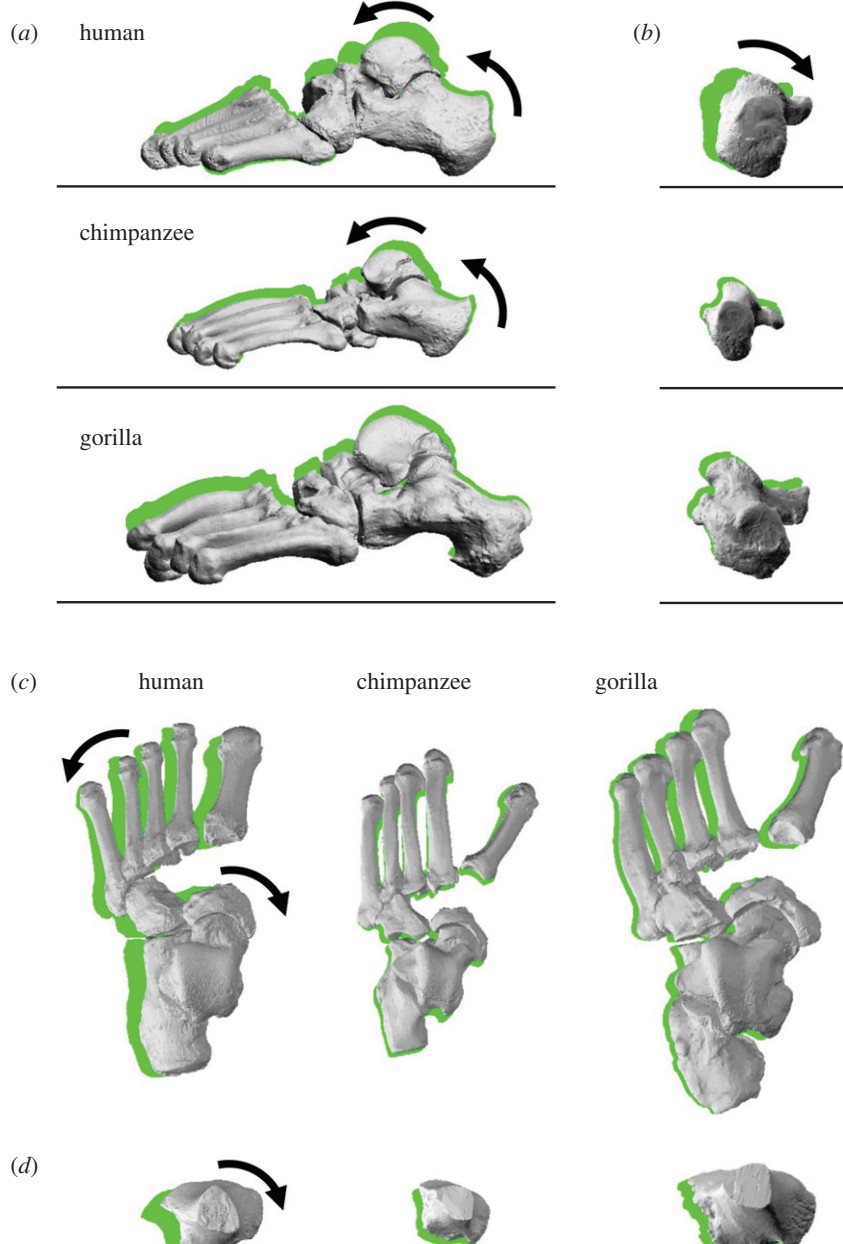

**Figure 2.** Comparisons of reconstructed three-dimensional foot bone movements under axial loading among humans, chimpanzees and gorillas. (*a*) Lateral view, (*b*) posterior view, (*c*) dorsal view of metatarsals, cuboid, navicular, talus and calcaneus and (*d*) dorsal view of tibia. The green shades indicate foot bone contours at the neutral posture.

## 2. Results

Figure 2 compares the 3D position and orientation of foot bones of representative human, chimpanzee and gorilla specimens before (0 N) and after the axial load (588 N) was applied. Figures 3–5 quantitatively compare the changes in the foot arch dimensions and tri-axial translational and rotational displacements of the foot bones with respect to the global coordinate system, respectively, from the neutral posture during axial loading. Figures 6 and 7 compare the changes in the tri-axial rotational displacements and joint angles, respectively, calculated using anatomical coordinate systems (see Material and methods). The results of the corresponding statistical tests between humans and African great apes are presented in tables 1–4. We lumped the two species of African apes for statistical tests because chimpanzees and gorillas are closer, if not identical, to one another than humans in terms of locomotion, and the present data indicated that the foot kinematics due to axial loading was generally more similar between the two species of African apes than between humans and African apes (see below).

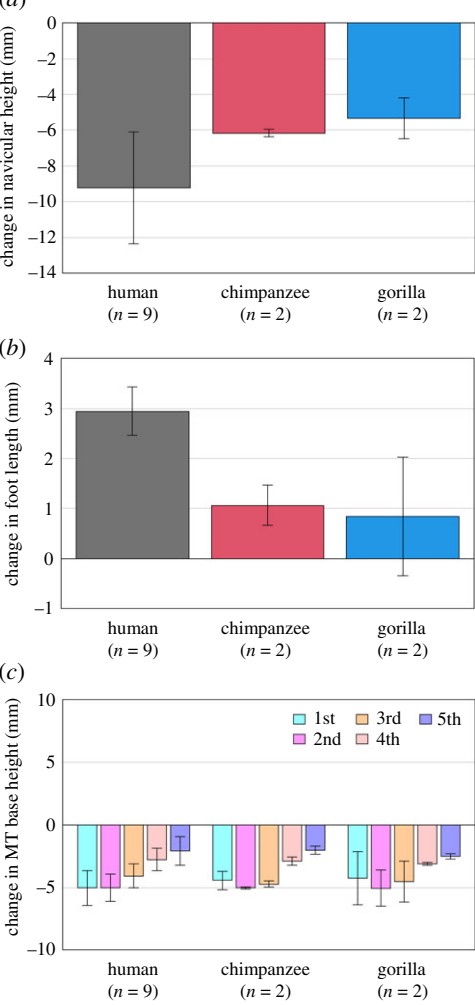

**Figure 3.** Comparisons of changes in the foot arch dimensions during axial loading among humans, chimpanzees and gorillas. Changes in (*a*) navicular height, (*b*) foot length and (*c*) MT base height from the neutral posture were quantified and compared. Bars and error bars indicate means and standard deviations, respectively.

**Table 1.** Changes in the foot arch dimensions during axial loading of humans and African great apes (chimpanzee + gorilla). *p*-values are presented if differences are significant.

|  | human | | African great apes | | |
|---|---|---|---|---|---|
|  | mean | s.d. | mean | s.d. | *p*-value |
| foot length[a] (mm) | 2.9 | 0.5 | 1.0 | 0.7 | $p = 0.001$ |
| navicular height[b] (mm) | −9.2 | 3.1 | −5.7 | 0.8 | $p = 0.017$ |
| MT base height (mm) |  |  |  |  |  |
| 1MT[b] | −5.0 | 1.4 | −4.4 | 1.3 |  |
| 2MT[b] | −5.0 | 1.1 | −5.0 | 0.8 |  |
| 3MT[b] | −4.1 | 1.0 | −4.6 | 1.0 |  |
| 4MT[b] | −2.8 | 0.9 | −3.0 | 0.2 |  |
| 5MT[b] | −2.1 | 1.1 | −2.3 | 0.4 |  |

[a]Alternative hypothesis is human > African great apes.
[b]Alternative hypothesis is human < African great apes.

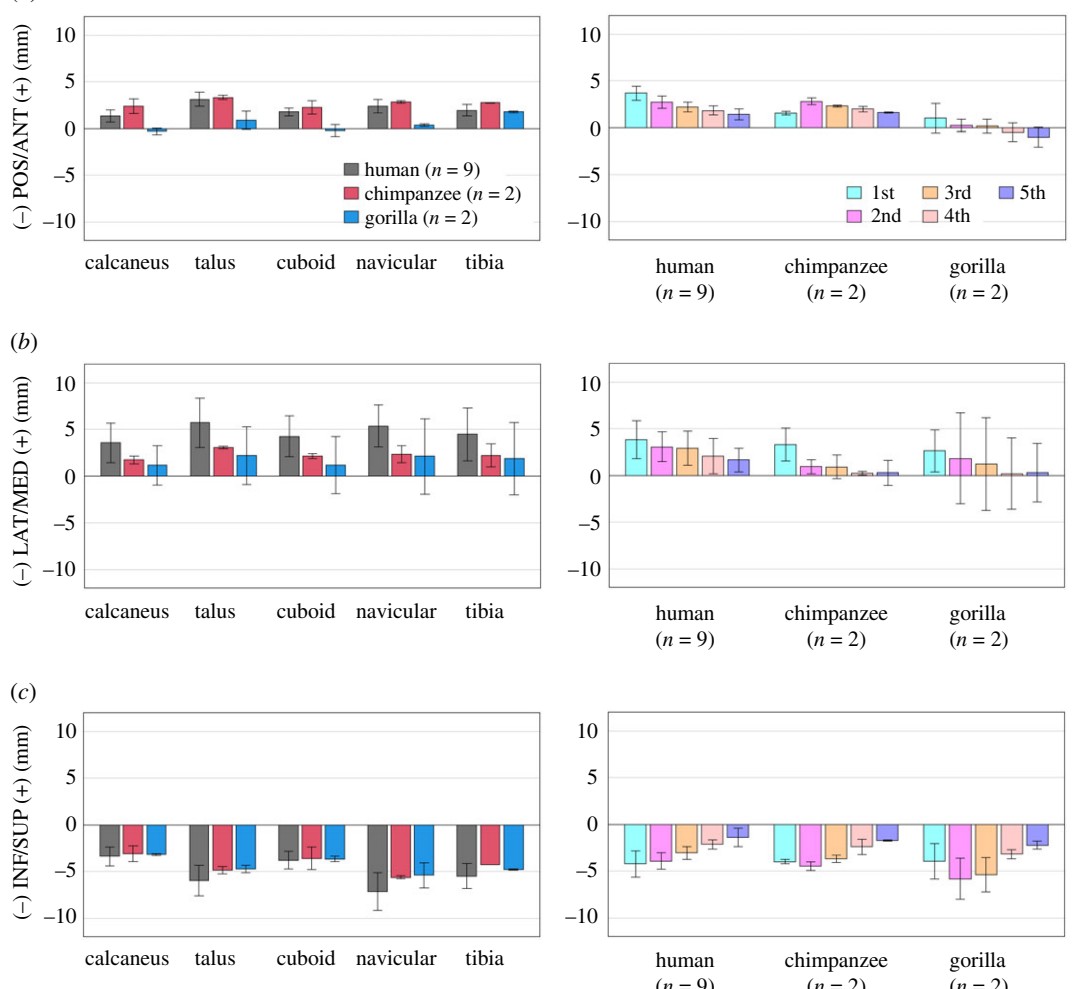

**Figure 4.** Comparisons of changes in the translational displacements of the foot bones during axial loading among humans, chimpanzees and gorillas. Translational displacements of the foot bones in the (*a*) anteroposterior, (*b*) mediolateral and (*c*) superoinferior directions from the neutral posture during axial loading were quantified and compared. The values are positive for anterior, medial and superior translation. Bars and error bars indicate means and standard deviations, respectively.

Due to axial loading, the height and length of the longitudinal arch decreased and increased, respectively, in human feet, but the values did not change largely in chimpanzee and gorilla feet (figure 3*a*,*b*), indicating that human feet are more deformable. However, the heights of the metatarsal (MT) bases decreased in a similar manner in all three species, indicating that the deformation pattern of the transverse arch was quite similar among the species (figure 3*c*).

During axial loading, all four tarsal bones and the tibia generally translated in the anterior, medial and inferior directions in humans and African great apes (figure 4*a*–*c*). However, medial translations of the calcaneus, talus, cuboid and navicular were significantly larger in humans than in African great apes. The MTs also translated in the same directions in all species (figure 4*a*–*c*). However, the anterior translation of the medial MTs and the medial translation of the lateral MTs tended to be larger in humans than in African great apes.

Owing to axial loading, the human calcaneus was significantly more everted than that of the African great apes (figures 5*a* and 6*a*). Inversion/eversion movement was not observed for the talus in all three species. The navicular was significantly more everted in humans than in chimpanzees and gorillas. The inversion/eversion movements of the MTs were generally in the everting direction in all three species but the 1MTs of the chimpanzees and gorillas were rotated in the inverting direction. Sagittally, the calcaneus and talus were rotated in the plantarflexing direction with respect to the global coordinate system in all three species, but the magnitudes were significantly larger in the human foot because of the flattening of the human foot (figure 5*b*). The MTs were dorsiflexed in all three species, but the dorsiflexion tended to be significantly larger in the human foot. Horizontally, the calcaneus, talus and tibia were generally rotated internally in all three species, except for the talus and tibia of the gorilla (figures 5*c* and 6*c*).

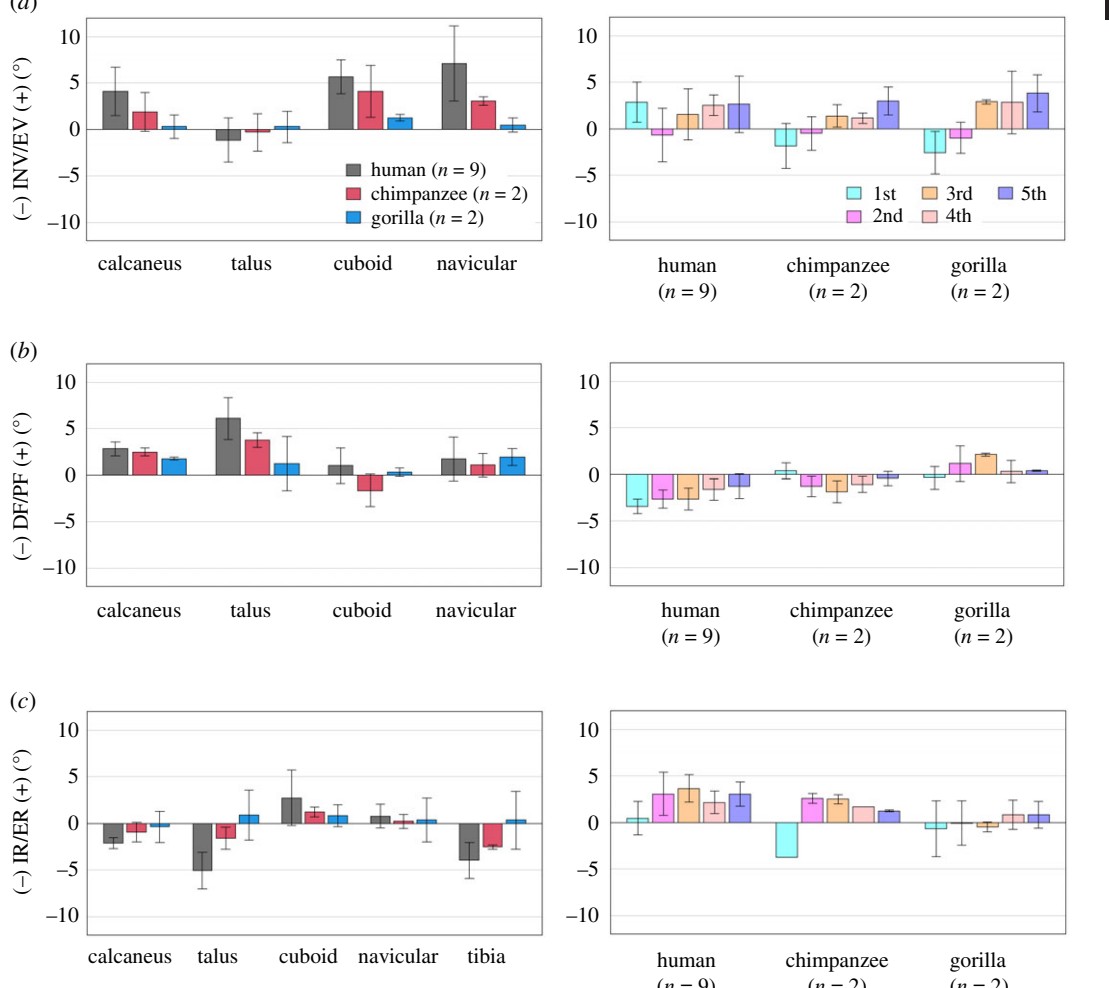

**Figure 5.** Comparisons of changes in the tri-axial rotations of the foot bones with respect to the global coordinate system during axial loading among humans, chimpanzees and gorillas. Rotational displacements of the foot bones in the (a) coronal, (b) sagittal and (c) transverse planes from the neutral posture during axial loading were quantified and compared. The values are positive for eversion, plantarflexion and external rotation. Bars and error bars indicate means and standard deviations, respectively.

The magnitudes of internal rotation of the talus and tibia were significantly larger in the human feet than in those of African great apes. The cuboid, navicular and MTs were all externally rotated in all three species, except for the 1MT of the African great apes. The magnitudes of external rotation of the MTs were generally larger in humans than in African great apes.

The talocalcaneal (TC) and talonavicular (TN) joints were found to be everted but the rotational angles of 1MT with respect to the navicular (N-1MT) was inverted due to axial loading in all three species (figure 7a). Among them, the human TC joint was significantly more everted than that of the African great apes (figure 7a). The dorsiflexions of the TC and N-1MT joints tended to be larger than those of the African apes, but the differences were not statistically significant (figure 7b). The external rotations of the human TC, TN and calcaneocuboid (CC) joints were significantly larger than those of the African apes (figure 7c). The external rotation of the TC joint means the internal rotation of the talus with respect to the calcaneus. Because of this, the navicular was calculated to be externally rotated with respect to the talus.

## 3. Discussion

During axial loading of the human foot, the calcaneus and tibia were everted and internally rotated, respectively, as documented previously [18–20]; however, we found that such coupling motion was only minimally observed in chimpanzees and gorillas (figures 5a,c and 6a,c, tables 2 and 3). In the

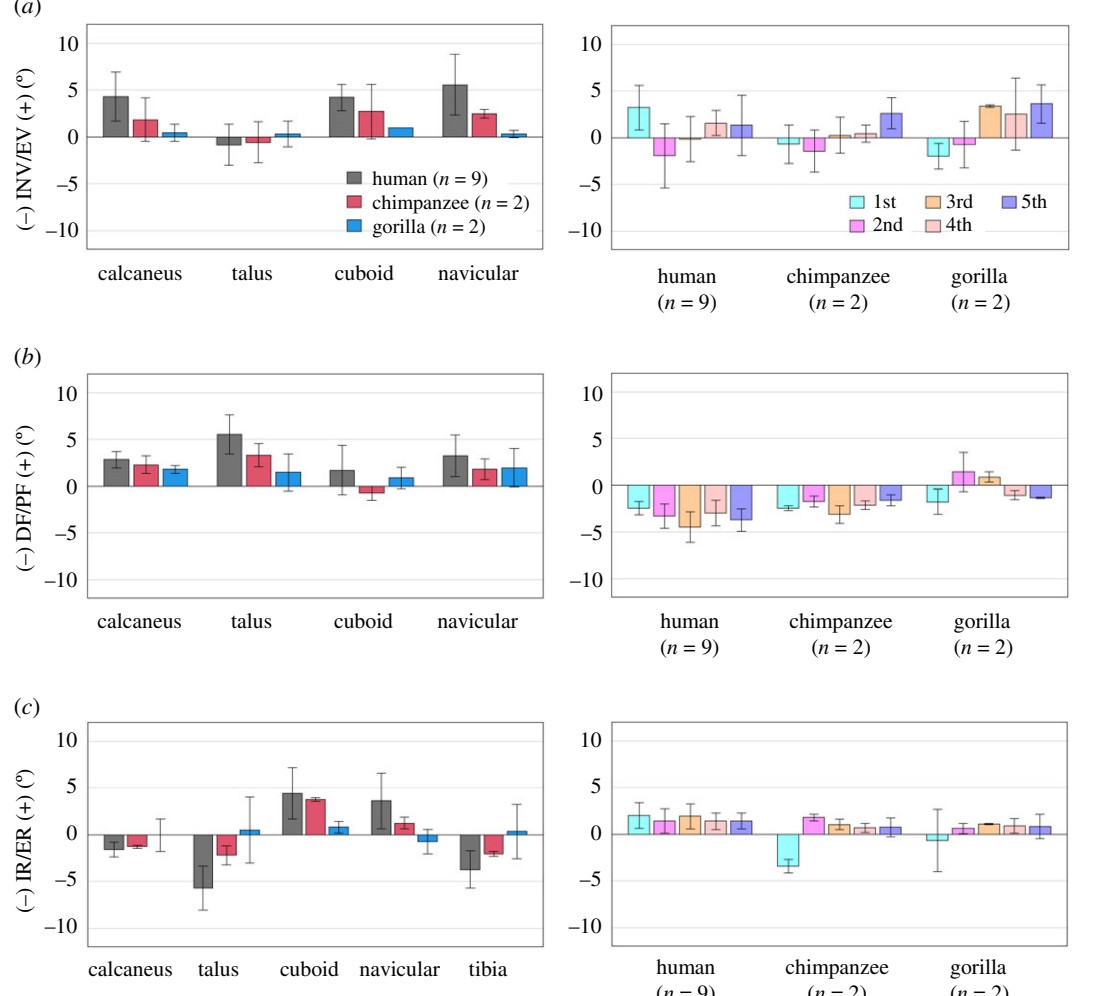

**Figure 6.** Comparisons of changes in the tri-axial rotations of the foot bones during axial loading among humans, chimpanzees and gorillas based on the anatomical coordinate systems. Rotational displacements of the foot bones in the (*a*) coronal, (*b*) sagittal and (*c*) transverse planes from the neutral posture during axial loading were quantified and compared. The values are positive for eversion, plantarflexion and external rotation. Bars and error bars indicate means and standard deviations, respectively.

human foot, the talus is internally rotated and slides medially down the subtalar articular surfaces of the calcaneus as the calcaneus is everted, resulting in the internal rotation of the tibia [18]. This coupling motion occurs mainly owing to the shape of the human calcaneus. In the human calcaneus, the lateral plantar process (LPP) is positioned plantarly [9], making the calcaneus tend to rotate in the everting direction when the foot is axially loaded (figure 8). On the other hand, the LPP is positioned more dorsally, and the calcaneal tuberosity is pointed more medioinferiorly in the posterior view in the feet of chimpanzees and gorillas (figure 8), indicating that the eversion of the calcaneus is relatively difficult, because the horizontal distance between the medial plantar process and the talus is shorter in the feet of the African great apes. As a consequence, the subtalar joint articular surface of the calcaneus is less tilted, and the medial translation and internal rotation of the talus are more restricted, possibly resulting in reduction of the internal rotation of the tibia in chimpanzees and gorillas. Furthermore, the present study demonstrated that the external rotation of the 2–5MTs was generally larger in humans than in African great apes. In the human foot, the talus was more internally rotated and the talar head moved more medially than in African great apes. As a result, the force applied to the MTs via the navicular during axial loading was directed more medially in the human foot, possibly resulting in the larger external rotations of the MTs in humans than in African great apes. Therefore, the innate mobility of the foot bones is clearly different between humans and African great apes. The rotational movements of the tibia and the MTs with respect to the vertical axis of the ground starting from the eversion of the calcaneus during axial loading, i.e. the tibio-calcaneal kinematic coupling, seem to be among the fundamental differences that distinguish the human foot from that of African great apes.

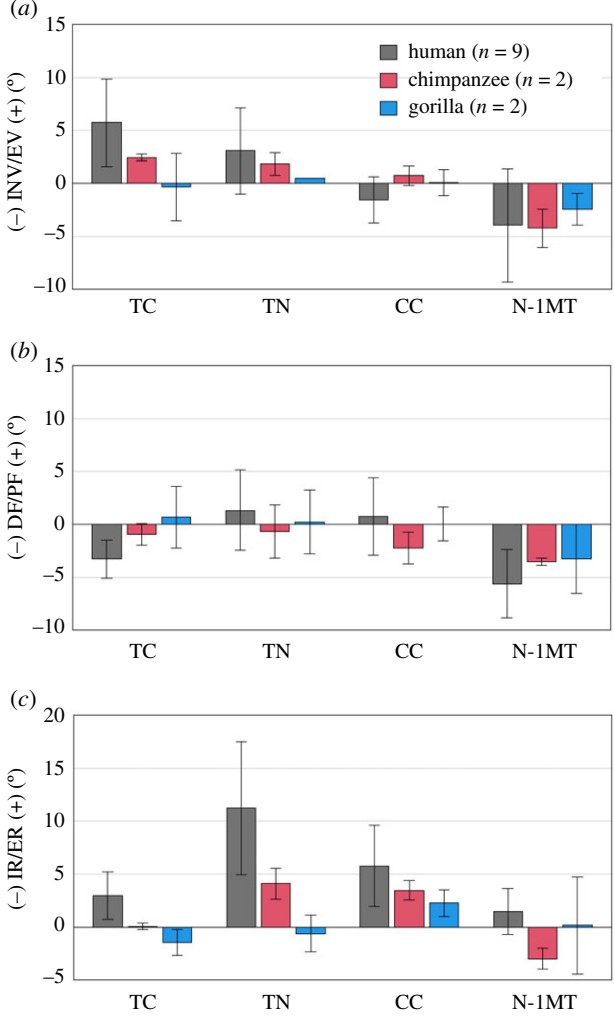

**Figure 7.** Comparisons of changes in talocalcaneal (TC), talonavicular (TN), calcaneocuboid (CC) joint angles and rotational angles of 1MT with respect to the navicular (N-1MT) during axial loading among humans, chimpanzees and gorillas. The values are positive for eversion, plantarflexion and external rotation. Bars and error bars indicate means and standard deviations, respectively.

During human walking and running, a vertical free moment (VFM), i.e. torque about the body's vertical axis due to friction between the foot and the ground, should be generated by the stance leg to counterbalance the axial moment due to trunk rotation and leg swing [22]. People on pointy stilts can walk without generating a VFM but it is more difficult to counterbalance the yaw moment generated by the swing leg and axial rotation of the trunk. Therefore, the above morphologically embedded mobility of the human foot may contribute functionally to structurally generate a VFM to the body during human bipedal locomotion. Actually, a recent cadaver study confirmed experimentally that axially loaded human feet possess a capacity to generate a VFM in the direction of internal rotation, the magnitude of which is quite comparable to that actually produced during human walking [34]. When a torque is applied to a deformable structure, it will twist along the axis of rotation. The greater the torque applied, the larger the rotation generated. The deformable human foot therefore behaves as a torsional spring when it is axially loaded. It has been suggested that the VFM is minimized during human walking as a result of arm-swinging to reduce the energetic cost of locomotion [23,25]. Therefore, the structurally embedded capacity of the human foot to generate a rotational movement around the yaw axis could be a functional adaptation to generation of efficient bipedal locomotion by facilitating the cancellation of the moment about the body's vertical axis during walking. How the innate structural mobility of the human foot actually contributes to the generation of the VFM and facilitates generation of bipedal locomotion is yet be clarified and should be examined in detail in future studies. However, the present study suggested that the tibio-calcaneal coupling motion during axial loading is possibly one of the derived characteristics of the human foot that is acquired as an adaptation to habitual bipedal locomotion in the human lineage.

**Table 2.** Changes in the tri-axial translational and rotational displacements of the foot bones with respect to the global coordinate system during axial loading of humans and African great apes (chimpanzee + gorilla). *p*-values are presented if differences are significant. The values are positive for anterior, medial and superior translations, and eversion, plantarflexion and external rotation.

| | human | | African great apes | | |
|---|---|---|---|---|---|
| | mean | s.d. | mean | s.d. | *p*-value |
| translation (mm) | | | | | |
| ANT/POST | | | | | |
| calcaneus[a] | 1.4 | 0.7 | 1.0 | 1.6 | |
| talus[a] | 3.1 | 0.8 | 2.1 | 1.5 | |
| cuboid[a] | 1.8 | 0.4 | 1.0 | 1.5 | |
| navicular[a] | 2.4 | 0.7 | 1.6 | 1.4 | |
| tibia[a] | 2.0 | 0.6 | 2.3 | 0.6 | |
| 1MT[a] | 3.7 | 0.8 | 1.3 | 1.0 | *p* = 0.001 |
| 2MT[a] | 2.7 | 0.6 | 1.5 | 1.5 | |
| 3MT[a] | 2.2 | 0.5 | 1.2 | 1.3 | |
| 4MT[a] | 1.8 | 0.5 | 0.8 | 1.5 | |
| 5MT[a] | 1.4 | 0.6 | 0.3 | 1.6 | |
| MED/LAT | | | | | |
| calcaneus[a] | 3.6 | 2.1 | 1.4 | 1.3 | *p* = 0.025 |
| talus[a] | 5.7 | 2.6 | 2.6 | 1.8 | *p* = 0.017 |
| cuboid[a] | 4.3 | 2.2 | 1.7 | 1.8 | *p* = 0.038 |
| navicular[a] | 5.4 | 2.3 | 2.2 | 2.4 | *p* = 0.025 |
| tibia[a] | 4.5 | 2.8 | 2.0 | 2.3 | |
| 1MT[a] | 3.8 | 2.0 | 3.0 | 1.7 | |
| 2MT[a] | 3.1 | 1.6 | 1.4 | 2.9 | |
| 3MT[a] | 2.9 | 1.8 | 1.1 | 3.0 | |
| 4MT[a] | 2.1 | 1.9 | 0.2 | 2.2 | |
| 5MT[a] | 1.7 | 1.3 | 0.3 | 2.0 | |
| SUP/INF | | | | | |
| calcaneus[b] | −3.4 | 1.0 | −3.1 | 0.5 | |
| talus[b] | −6.0 | 1.6 | −4.8 | 0.3 | |
| cuboid[b] | −3.8 | 0.9 | −3.6 | 0.7 | |
| navicular[b] | −7.1 | 2.0 | −5.5 | 0.8 | |
| tibia[b] | −5.5 | 1.3 | −4.5 | 0.3 | |
| 1MT[b] | −4.2 | 1.4 | −4.0 | 1.1 | |
| 2MT[b] | −3.9 | 0.9 | −5.1 | 1.5 | |
| 3MT[b] | −3.0 | 0.7 | −4.5 | 1.5 | |
| 4MT[b] | −2.1 | 0.5 | −2.8 | 0.7 | |
| 5MT[b] | −1.4 | 1.0 | −2.0 | 0.4 | |
| rotation (°) | | | | | |
| EV/INV | | | | | |
| calcaneus[a] | 4.1 | 2.6 | 1.1 | 1.7 | *p* = 0.038 |
| talus[a] | −1.1 | 2.4 | 0.0 | 1.5 | |

(*Continued.*)

| | human | | African great apes | | |
|---|---|---|---|---|---|
| | mean | s.d. | mean | s.d. | *p*-value |
| cuboid[a] | 5.7 | 1.8 | 2.7 | 2.3 | p = 0.025 |
| navicular[a] | 7.1 | 4.0 | 1.8 | 1.6 | p = 0.025 |
| 1MT[a] | 2.9 | 2.1 | −2.2 | 2.0 | p = 0.003 |
| 2MT[a] | −0.7 | 2.9 | −0.7 | 1.4 | |
| 3MT[a] | 1.5 | 2.7 | 2.1 | 1.1 | |
| 4MT[a] | 2.5 | 1.1 | 2.0 | 2.2 | |
| 5MT[a] | 2.6 | 3.0 | 3.4 | 1.5 | |
| PF/DF | | | | | |
| calcaneus[a] | 2.8 | 0.7 | 2.1 | 0.5 | p = 0.038 |
| talus[a] | 6.1 | 2.3 | 2.5 | 2.3 | p = 0.017 |
| cuboid[a] | 1.0 | 1.9 | −0.7 | 1.5 | |
| navicular[a] | 1.8 | 2.3 | 1.5 | 1.0 | |
| 1MT[b] | −3.4 | 0.8 | 0.0 | 0.9 | p = 0.001 |
| 2MT[b] | −2.6 | 1.0 | −0.1 | 1.9 | p = 0.010 |
| 3MT[b] | −2.6 | 1.2 | 0.1 | 2.4 | p = 0.025 |
| 4MT[b] | −1.6 | 1.1 | −0.4 | 1.1 | |
| 5MT[b] | −1.3 | 1.3 | 0.0 | 0.6 | p = 0.038 |
| ER/IR | | | | | |
| calcaneus[b] | −2.1 | 0.6 | −0.7 | 1.2 | p = 0.006 |
| talus[b] | −5.1 | 2.0 | −0.3 | 2.2 | p = 0.006 |
| cuboid[b] | 2.7 | 3.0 | 1.0 | 0.8 | |
| navicular[b] | 0.8 | 1.3 | 0.3 | 1.4 | |
| tibia[b] | −4.0 | 1.9 | −1.1 | 2.4 | p = 0.017 |
| 1MT[a] | 0.5 | 1.8 | −2.2 | 2.5 | |
| 2MT[a] | 3.1 | 2.3 | 1.3 | 2.1 | |
| 3MT[a] | 3.7 | 1.5 | 1.0 | 1.8 | p = 0.017 |
| 4MT[a] | 2.2 | 1.2 | 1.3 | 1.0 | |
| 5MT[a] | 3.0 | 1.3 | 1.0 | 0.9 | p = 0.010 |

[a]Alternative hypothesis is human > African great apes.
[b]Alternative hypothesis is human < African great apes.

Another important finding of the present study is that the changes in the length of the longitudinal arch due to axial loading were larger in the human foot than in the feet of chimpanzees and gorillas, indicating that the human foot is more deformable in the sagittal plane. This is due to the arch-like structure of the human foot in which the midfoot bones are elevated by the well-developed plantar aponeurosis connecting from the calcaneal tuber to the bases of the proximal phalanges. Therefore, the human foot has a larger capacity to be flattened due to axial loading, and the plantar aponeurosis is stretched to store elastic energy [35,36]. On the other hand, the structure of chimpanzee and gorilla feet is more flattened in an unloaded neutral condition. As a consequence, the feet were less deformable in chimpanzees and gorillas during axial loading in the present study. This observation corresponds to a recent experimental study of midfoot motion during bipedal walking in humans and chimpanzees [30] reporting that the range of the midfoot joint motion in the stance phase is much larger in humans than in chimpanzees. It has been generally accepted that the human foot is less compliant than that of African great apes [12], and this has been considered a main factor for the

**Table 3.** Changes in the tri-axial rotational displacements of the foot bones during axial loading of humans and African great apes (chimpanzee + gorilla) calculated based on the anatomical coordinate systems. p-values are presented if differences are significant. The values are positive for eversion, plantarflexion and external rotation.

| | human | | African great apes | | |
|---|---|---|---|---|---|
| | mean | s.d. | mean | s.d. | *p*-value |
| rotation (°) | | | | | |
| EV/INV | | | | | |
| calcaneus[a] | 4.3 | 2.6 | 1.1 | 1.6 | *p* = 0.038 |
| talus[a] | −0.8 | 2.2 | −0.1 | 1.6 | |
| cuboid[a] | 4.2 | 1.4 | 1.8 | 2.0 | *p* = 0.025 |
| navicular[a] | 5.6 | 3.3 | 1.4 | 1.3 | *p* = 0.017 |
| 1MT[a] | 3.2 | 2.4 | −1.3 | 1.6 | *p* = 0.006 |
| 2MT[a] | −1.9 | 3.4 | −1.1 | 2.0 | |
| 3MT[a] | −0.1 | 2.4 | 1.8 | 2.1 | |
| 4MT[a] | 1.6 | 1.4 | 1.5 | 2.6 | |
| 5MT[a] | 1.3 | 3.3 | 3.1 | 1.6 | |
| PF/DF | | | | | |
| calcaneus[a] | 2.8 | 0.9 | 2.0 | 0.7 | |
| talus[a] | 5.5 | 2.1 | 2.4 | 1.7 | *p* = 0.017 |
| cuboid[a] | 1.7 | 2.7 | 0.1 | 1.2 | |
| navicular[a] | 3.2 | 2.2 | 1.9 | 1.4 | |
| 1MT[b] | −2.4 | 0.7 | −2.1 | 0.9 | |
| 2MT[b] | −3.3 | 1.3 | −0.2 | 2.2 | *p* = 0.003 |
| 3MT[b] | −4.5 | 1.6 | −1.1 | 2.4 | *p* = 0.017 |
| 4MT[b] | −3.0 | 1.4 | −1.6 | 0.7 | *p* = 0.010 |
| 5MT[b] | −3.7 | 1.2 | −1.4 | 0.4 | *p* = 0.003 |
| ER/IR | | | | | |
| calcaneus[b] | −1.6 | 0.8 | −0.7 | 1.2 | |
| talus[b] | −5.7 | 2.4 | −0.9 | 2.6 | *p* = 0.010 |
| cuboid[b] | 4.4 | 2.7 | 2.3 | 1.7 | |
| navicular[b] | 3.6 | 3.0 | 0.3 | 1.4 | |
| tibia[b] | −3.7 | 2.0 | −0.9 | 2.2 | *p* = 0.010 |
| 1MT[a] | 2.0 | 1.4 | −2.1 | 2.5 | *p* = 0.038 |
| 2MT[a] | 1.4 | 1.3 | 1.2 | 0.8 | |
| 3MT[a] | 1.9 | 1.3 | 1.1 | 0.3 | |
| 4MT[a] | 1.4 | 0.9 | 0.8 | 0.6 | |
| 5MT[a] | 1.4 | 0.9 | 0.8 | 1.0 | |

[a]Alternative hypothesis is human > African great apes.
[b]Alternative hypothesis is human < African great apes.

presence of the midtarsal break in bipedal locomotion in chimpanzees [4,11,37]. However, because the human foot has a deformable, arch-like structure, elastic energy storage and release during locomotion become possible in the human foot. The midfoot of the human foot is stiff during push-off because of the so-called Windlass mechanism [7,38], in which the plantar aponeurosis pulled by the dorsiflexion of the metatarsophalangeal joints develops the tensile force that can encumber the dorsiflexion of the midfoot joint. In chimpanzee and gorilla feet, on the other hand, the plantar aponeurosis was not as

**Table 4.** Changes in talocalcaneal (TC), talonavicular (TN), calcaneocuboid (CC) joints and rotational angles of the first metatarsal with respect to navicular (N-1MT) during axial loading of humans and African great apes (chimpanzee + gorilla). *p*-values are presented if differences are significant. The values are positive for eversion, plantarflexion and external rotation.

| | human | | African great apes | | |
|---|---|---|---|---|---|
| | mean | s.d. | mean | s.d. | *p*-value |
| rotation (°) | | | | | |
| EV/INV | | | | | |
| TC[a] | 5.7 | 4.1 | 1.0 | 2.4 | *p* = 0.010 |
| TN[a] | 3.1 | 4.0 | 1.2 | 1.0 | |
| CC[a] | −1.5 | 2.2 | 0.4 | 1.0 | |
| N-1MT[a] | −3.9 | 5.3 | −3.3 | 1.7 | |
| PF/DF | | | | | |
| TC[a] | −3.3 | 1.8 | −0.1 | 2.0 | |
| TN[a] | 1.3 | 3.8 | −0.2 | 2.3 | |
| CC[a] | 0.7 | 3.6 | −1.1 | 1.8 | |
| N-1MT[a] | −5.6 | 3.2 | −3.4 | 1.9 | |
| ER/IR | | | | | |
| TC[a] | 3.0 | 2.2 | −0.7 | 1.1 | *p* = 0.010 |
| TN[a] | 11.2 | 6.3 | 1.8 | 3.0 | *p* = 0.010 |
| CC[a] | 5.8 | 3.8 | 2.9 | 1.1 | *p* = 0.025 |
| N-1MT[a] | 1.5 | 2.2 | −1.4 | 3.3 | |

[a]Alternative hypothesis is human > African great apes.
[b]Alternative hypothesis is human < African great apes.

well developed as in the human foot [4,38,39], and hence the feet are possibly more compliant during push-off in chimpanzees and gorillas. It must be noted, however, recent studies suggested that the increase in the stiffness of the foot is more attributable to plantar intrinsic foot muscle activation [40–43]. Further studies are necessary to clarify the detailed mechanism underlying the increased stiffness of the foot enhancing arch recoil at push-off in human bipedal locomotion.

It has been generally accepted that the CC joint is relatively more locked in humans than in the African apes because a prominent medial process of the cuboid is present in the human foot, and it articulates with a deep depression on the distomedial surface of the calcaneus. Contrary to this concept, the present study suggested the CC joint in the human foot was comparatively more mobile in the direction of external rotation than in the feet of African apes (figure 7), possibly due to the fact that the human foot is actually more deformable as observed in the present study. On the other hand, in the gorilla foot, the CC joint was comparatively more immobile during axial loading. The CC joint surface morphology is known to differ between chimpanzees and gorillas and is reportedly flatter in gorillas [44,45]. Such a difference in the joint morphology might explain the difference in the mobility of the CC joint among the three species. However, the locking mechanism of the human foot is more important during push-off, but not during axial loading. The difference in the innate mobility of the foot bones between humans and African great apes at the time of push-off should also be investigated in future studies.

In conclusion, the present study clarified the differences in the foot bone mobility during axial loading between humans and African great apes and found that the morphologically embedded tibio-calcaneal kinematic coupling observed in the human foot is probably one of the derived features of the human foot that possibly facilitate energy-efficient bipedal locomotion. However, the present study has several limitations. The first limitation of the present study is that we quantified the foot bone movement from only two chimpanzees and two gorillas, even though the foot bone mobility could be highly variable. Therefore, the studied specimens may not have been representatives of each species. Although the availability of fresh frozen specimens is extremely limited for chimpanzees and

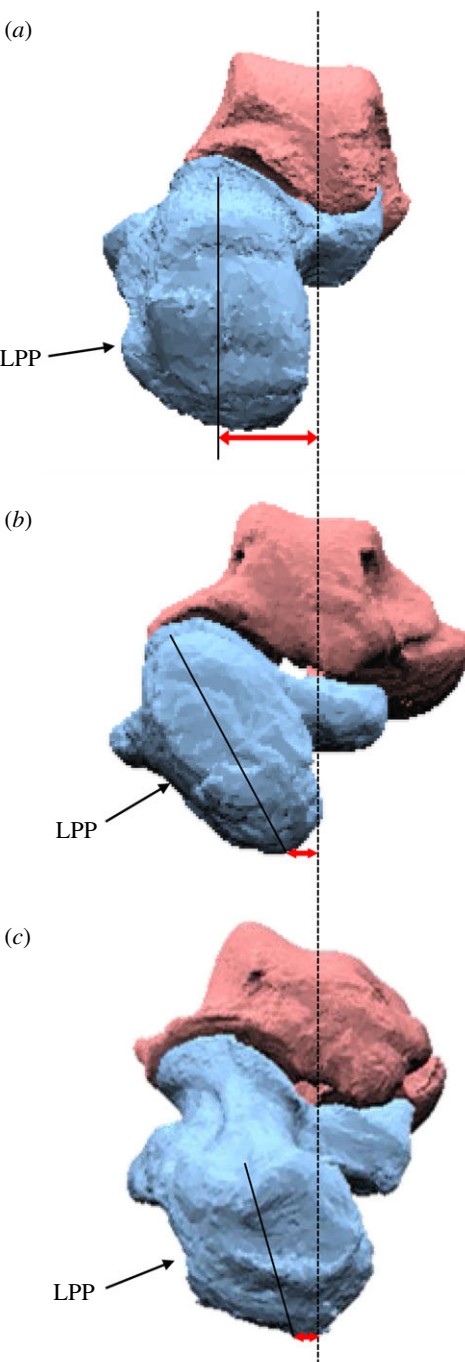

**Figure 8.** Posterior view of the (*a*) human, (*b*) chimpanzee and (*c*) gorilla calcanei (left). The lateral plantar process (LPP) of the human calcaneus is positioned plantarly. On the other hand, the LPP is positioned more dorsally, and the calcaneal tuberosity is pointed more medioinferiorly in the feet of chimpanzees and gorillas. Because the horizontal distance between the medial plantar process and the talus is relatively longer in the human foot than in the feet of the African great apes, the human calcaneus tends to easily rotate in the everting direction when the foot is axially loaded.

gorillas, the present results should be confirmed by examining a larger number of cases when chances arise in future studies. Second, since the aim of the current study was to extract the differences in the innate mobility of the foot bones during axial loading among three species, the present study only deals with a static analysis of the foot, but the present static analysis alone does not fully explain the functional significance of the tibio-calcaneal coupling embedded in the human foot on generation of human bipedal locomotion. In addition, the present study did not consider the contributions of foot muscles activations which possibly alter the mechanical interaction of the feet with the ground.

How the morphologically embedded tibio-calcaneal kinematic coupling could possibly contribute to the whole-body dynamics and energetics of human bipedal locomotion *in vivo* should be investigated in future studies.

# 4. Material and methods

These methods expand upon those detailed within our previous work [18].

## 4.1. Specimens

Fresh frozen cadaver lower legs of nine humans (mean age at death, 79.4 years; range, 63–90 years; six males and three females), two chimpanzees (30 and 24 years old; two males) and two gorillas (38 and 34 years old; two males) were obtained for X-ray measurements. All specimens were confirmed to be free of foot and ankle pathologies by visual and radiographic inspection. The chimpanzee and gorilla specimens were donated to the Primate Research Institute, Kyoto University by Himeji Central Park, Fukuoka City Zoological Garden, and Kobe Oji Zoo. The human specimens were donated to Keio University School of Medicine with the informed consent of the families of all of the donors. The kinematic data of five humans were taken from our previous study [18], but the rest of the data were newly obtained in the present study. The study on the human cadaver feet was approved by the ethics committee of the Faculty of Science and Technology and School of Medicine, Keio University. All methods were performed in accordance with the relevant guidelines and regulations.

The human specimens were cut at the middle of the shank prior to the experiment. For the chimpanzee and gorilla specimens, complete lower legs with intact tibia and fibula were obtained, as it was not allowed to cut the bones because the great ape specimens needed to be preserved for preparation of dry skeletons after the experiment. Extrinsic foot muscles were completely stripped and removed from the shafts of the tibia and fibula. Therefore, although the pedal phalanges are naturally flexed when the ankle is dorsiflexed in intact feet of chimpanzees and gorillas owing to the relatively short tendons of their pedal flexor muscles, the pedal phalanges could be completely extended even if the ankle joint was dorsiflexed in the current preparation of the specimens.

## 4.2. Experimental set-up

Weights of 5 kg were placed up to 60 kg (588 N) on the shafts affixed to cadaver feet for vertical loading. A biplanar fluoroscopy system was then employed to observe 3D kinematics of the foot bones during axial loading (figure 1a). We created a 3D-printed socket made of rubber-like polymer material to affix the proximal ends of the amputated tibia and fibula to the shaft of the human specimens to align the long axes of the lower leg and shaft. The long axis of the lower leg of the human specimen was defined as the line connecting the midpoint of the centroids of the tibial and fibular cross-sections and the midpoint between the lateral and medial malleoli. In the case of the great ape specimens, the entire proximal end of the tibia and fibula including the articular surface of the tibia, the tibial tuberosity and the fibular head was covered by the socket for fixation. The long axis of the lower leg of the chimpanzees and gorillas was defined as the line connecting the midpoint between the medial and lateral tibial condyles and the midpoint between the medial and lateral malleoli. See Ito *et al*. [18] for details of the fixation of the specimens to the shaft using the custom-made socket.

The shaft goes through a linear motion rolling guide fixed to the X-ray system to move freely along and rotate around the vertical axis. This permits unrestrained tibial rotation, which is a possible result of tibio-calcaneal coupling generated during axial loading. Each foot was placed at plantigrade on the acrylic table with a thin rubber sheet affixed on the surface to prevent slippage during axial loading. The long axis of the foot, defined by the line connecting the calcaneal tuberosity and the head of 2MT, was aligned with the anteroposterior axis of the global coordinate system. The pedal phalanges of the chimpanzees and gorillas were extended and placed naturally along the floor. Placement of chimpanzee and gorilla specimens at complete plantigrade is impossible due to slight inversion of the plantar surface of the foot in these species; the inverted foot is one of the intrinsic morphological differences between the human and African ape feet. However, to clarify the differences in the foot bone motions owing to such differences in the foot morphology, we tried to place the foot as neutrally as possible so that excessive ankle eversion did not occur at its initial, zero-loading condition. The

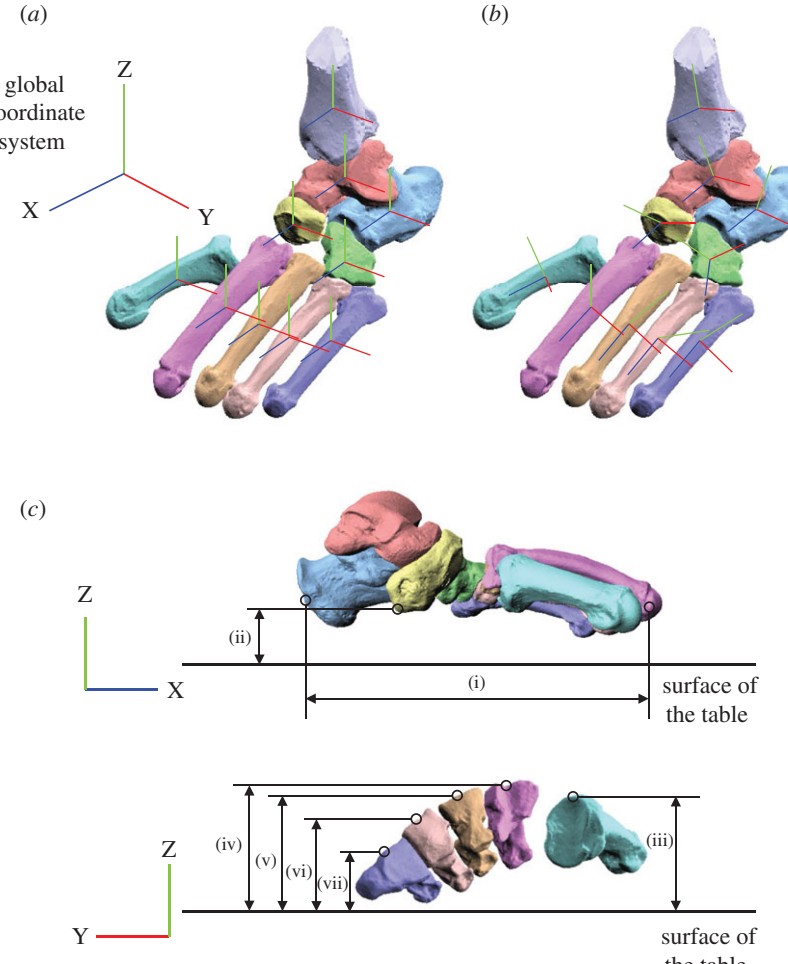

**Figure 9.** Definition of the coordinate systems (*a,b*) and foot dimensions (*c*). We defined bone-fixed local coordinate systems (*a*) such that the three orthogonal axes of the local coordinate system at the neutral posture (xyz) were aligned with the global coordinate system (XYZ), and (*b*) using anatomical landmarks defined in figure 10. The xyz axes (blue, red, green, respectively) approximately point anteriorly, medially and superiorly, respectively. Foot arch dimensions were (i) foot length and (ii) navicular height as measures of the longitudinal arch and the (iii–vii) MT base height of 1–5MTs as measures of the transverse arch.

hallux was placed in the neutral position, approximately 30° with respect to the long axis of the foot. To compare the innate mobility of the foot bones produced just by the morphology and structure of the foot, no tendon tractions were applied in the present study.

## 4.3. Biplane X-ray fluoroscopy and model-based registration

A biplanar X-ray fluoroscopy system was used to observe the resulting foot bony movement of axial loading [46]. The left and right fluoroscopic images were captured at every 10 kg of applied load. To allow the foot bones to rest after loading, we waited for at least 30 s before obtaining the next images. Measurement was taken once for each specimen.

We used a model-based matching method to quantify 3D foot bone kinematics under axial loading [46]. Bone tomography was computed and then used to create 3D bone surface models of the cadaver foot before our experiment. Only the distal one-fourth of the tibia was used to construct the corresponding surface model. Spatial calibration of the biplanar fluoroscopic system reconstructed the projection geometry of the system. Three-dimensional surface models were matched to the fluoroscopic images to maximize the similarity of occluding contours of bone surface models with the corresponding edge-enhanced fluoroscopic images (figure 1*b,c*) using a quasi-Newton method, while avoiding mutual penetration of bones. The accuracy of the bone registration was reportedly approximately 0.3 mm and 0.3° for translation and orientation, respectively [41].

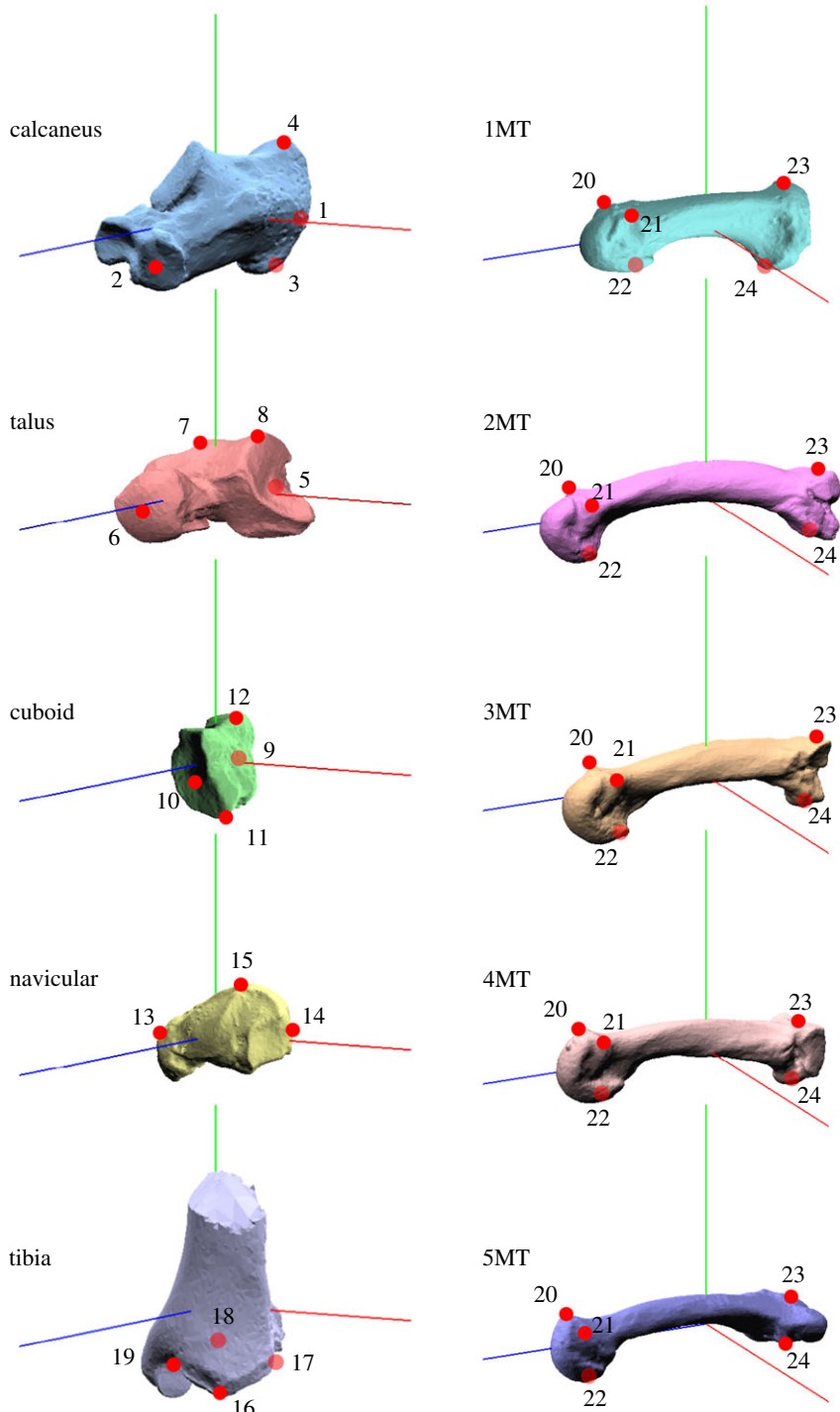

**Figure 10.** Landmarks used in the present study. See Gutekunst et al. [49] for the definitions of the landmarks on the calcaneus, talus, cuboid and navicular. Landmarks 16, 17, 18 and 19 on the tibia = anterolateral, posterolateral, posteromedial and anteromedial edges of the plafond, respectively; landmarks 20, 21, 22, 23 and 24 on each MT = most medial and lateral points on the dorsal border of MT head, midpoint of the plantar edge of MT head, most dorsal point on the dorsal border and most plantar point on the plantar border of the proximal articular surface, respectively.

## 4.4. Quantifications of bone movements

To quantitatively compare the 3D translations and rotations of the nine foot bones, i.e. the calcaneus, talus, navicular, cuboid, and five MTs, and the tibia with respect to the global coordinate system due to axial loading among humans, chimpanzees and gorillas, we defined a local coordinate system fixed to each bone such that the three orthonormal axes of the local coordinate system at the neutral

**Table 5.** Bone coordinate system definitions based on anatomical landmarks (left foot). $m(a,b)$ = midpoint between $a$ and $b$.

| bone | first axis | temporary axis | second axis | third axis |
|---|---|---|---|---|
| calcaneus | $\mathbf{X}_{CAL} = \lvert 1 \to 2 \rvert$ | $\mathbf{t}_{CAL} = \lvert 3 \to 4 \rvert$ | $\mathbf{Y}_{CAL} = \mathbf{t}_{CAL} \times \mathbf{X}_{CAL}$ | $\mathbf{Z}_{CAL} = \mathbf{X}_{CAL} \times \mathbf{Y}_{CAL}$ |
| talus | $\mathbf{X}_{TAL} = \lvert 5 \to 6 \rvert$ | $\mathbf{t}_{TAL} = \lvert 7 \to 8 \rvert$ | $\mathbf{Z}_{TAL} = \mathbf{X}_{TAL} \times \mathbf{t}_{TAL}$ | $\mathbf{Y}_{TAL} = \mathbf{Z}_{TAL} \times \mathbf{X}_{TAL}$ |
| cuboid | $\mathbf{X}_{CUB} = \lvert 9 \to 10 \rvert$ | $\mathbf{t}_{CUB} = \lvert 11 \to 12 \rvert$ | $\mathbf{Y}_{CUB} = \mathbf{t}_{CUB} \times \mathbf{X}_{CUB}$ | $\mathbf{Z}_{CUB} = \mathbf{X}_{CUB} \times \mathbf{Y}_{CUB}$ |
| navicular | $\mathbf{Y}_{NAV} = \lvert 13 \to 14 \rvert$ | $\mathbf{t}_{NAV} = \lvert 13 \to 15 \rvert$ | $\mathbf{X}_{NAV} = \mathbf{Y}_{NAV} \times \mathbf{t}_{NAV}$ | $\mathbf{Z}_{NAV} = \mathbf{X}_{NAV} \times \mathbf{Y}_{NAV}$ |
| tibia | $\mathbf{Y}_{TIB} = \lvert m(18,19) \to m(16,17) \rvert$ | $\mathbf{t}_{TIB} = \lvert m(17,18) \to m(16,19) \rvert$ | $\mathbf{Z}_{TIB} = \mathbf{t}_{TIB} \times \mathbf{Y}_{TIB}$ | $\mathbf{X}_{TIB} = \mathbf{Y}_{TIB} \times \mathbf{Z}_{TIB}$ |
| MT | $\mathbf{X}_{MT} = \lvert m(23,24) \to m(m(20,21),22) \rvert$ | $\mathbf{t}_{MT} = \lvert 21 \to 23 \rvert \times \lvert 21 \to 20 \rvert$ | $\mathbf{Y}_{MT} = \mathbf{t}_{MT} \times \mathbf{X}_{MT}$ | $\mathbf{Z}_{MT} = \mathbf{X}_{MT} \times \mathbf{Y}_{MT}$ |

posture (xyz) were aligned with those of the global coordinate system (XYZ) (figure 9a) [47,48]. In this study, we manually placed each specimen to align the long axis of the foot to the anteroposterior axis of the global coordinate system. The rotations around the X, Y and Z axes were referred in the present study as inversion–eversion (supination–pronation), plantarflexion–dorsiflexion and internal/external rotation (adduction–abduction), respectively. The origin of the bone coordinate system was defined by the centroid of the corresponding bone. The centroid of the bone model was calculated by averaging each coordinate of the points on the mesh surface model. The posture when only the vertical shaft (3.3 kg) was affixed to the specimen (zero-loading condition) was defined as the neutral posture. We quantified the change in the position and orientation of the foot bones from the neutral posture using y-x-z Euler angles (figure 9a). The mean and standard deviation of the arch measurements and the translational and rotational displacement of the foot bones were calculated for each species for comparisons.

In addition, we also compared the 3D rotations of the nine foot bones by defining an anatomical local coordinate system to each bone (figure 9b) based on bony landmarks (figure 10). The local coordinate systems of the calcaneus, talus, cuboid and navicular were defined by referring to Gutekunst et al. [49]. Those of the tibia and five metatarsals were calculated using four or five anatomical landmarks digitized on each of the bones (figure 10). Computations of the three orthogonal axes for each bone coordinate system using the anatomical landmarks are provided in table 5. We quantified the change in the orientation of the foot bones from the neutral posture using y-x-z Euler angles. Furthermore, the changes in the TC, TN, CC and N-1MT joint angles were calculated using the anatomical coordinate systems. The relative rotation of the distal bone with respect to the proximal bone was quantified using y-x-z Euler angles.

To compare the macroscopic deformation characteristics of the foot during axial loading, we calculated the changes in the length and height of the longitudinal arch (figure 9c). The length of the longitudinal arch was defined as the sagittally projected length of the distance between the most posterior point of the calcaneal tuberosity and the most distal point of the medial edge of the 2MT head articular surface. The navicular height (the vertical position of the navicular tuberosity from the surface of acrylic table) was calculated as a measure of the height of longitudinal arch. To quantify the deformation of the transverse arch, the changes in the height of base of MTs were also calculated (figure 9c). In this study, the change in the vertical position of the midpoint of dorsal edge of the MT proximal articular surface was calculated.

## 4.5. Statistical analysis

Since the number of African great ape specimens is quite limited, and the kinematic characteristics of the chimpanzee and gorilla feet are quite similar to each other, the changes due to axial loading in the linear measurement and the spatial positions and orientations of the foot bones and tibia were statistically compared between humans and African great apes (chimpanzees + gorillas). One-tailed two-sample Mann–Whitney U-tests were performed using R v. 4.0.4 software [50] to assess statistical differences of the mean values between humans and African great apes. The alternative hypothesis here is that the translations and orientations of the foot bones are larger in magnitude in humans than in African great apes.

Ethics. The study on the human cadaver feet was approved by the ethics committee of the Faculty of Science and Technology and School of Medicine, Keio University. All methods were performed in accordance with the relevant guidelines and regulations.

Data accessibility. The raw data of figures 3–7 and the matrices representing the bone coordinate systems before and after axial loading are provided as electronic supplementary material [51].

Authors' contributions. T.N., K.I., K.H. and N.O. conceived and designed the study. T.N., K.I., T.N., T.O., N.I., M.J., M.O. and N.O. performed the experiments. T.N., K.I. and N.O. performed data analysis and drafted the manuscript, and all authors edited and approved the manuscript prior to submission.

Competing interests. The authors declare no competing interests.

Funding. This study was financially supported by a Grant-in-Aid for Scientific Research (grant nos. 17H01452, 23220004, 20H03331) from the Japan Society for the Promotion of Science, and the Cooperation Research Program of the Primate Research Institute, Kyoto University.

Acknowledgements. The authors express their sincere gratitude to the Clinical Anatomy Laboratory, School of Medicine, Keio University for approving the use of the laboratory for cadaver studies and Shimadzu Corporation, Kyoto, Japan, for the development of the biplanar X-ray fluoroscopy system used in the present study. The authors also thank all the staff of Himeji Central Park, Fukuoka City Zoological Garden, Kobe Oji Zoo, the Great Ape Information Network and the Primate Research Institute of Kyoto University for kindly allowing us to use the great ape specimens.

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
