## [Peer Review File · Royal Society Open Science]

Review History

RSOS-210903.R0 (Original submission)

Review form: Reviewer 1 (Kristiaan D'Août)

Is the manuscript scientifically sound in its present form?

Yes

Are the interpretations and conclusions justified by the results?

Yes

Is the language acceptable?

Yes

Do you have any ethical concerns with this paper?

No

Have you any concerns about statistical analyses in this paper?

No

Recommendation?

Accept with minor revision (please list in comments)

Comments to the Author(s)

This manuscript presents new data on foot bone mobility in two chimpanzees, two gorillas, and four humans, and includes data on five more humans published previously.

This manuscript has straightforward results and is quite descriptive, but the kinematic differences between species have been interpreted well and linked to the overall anatomy.

The methods are well described.

I believe this paper will provide a valuable contribution to the literature.

For me, the most interesting message is that the human foot is actually the more mobile one, which goes against “textbook” knowledge that human feet are stiff and non-human primate feet are flexible. This message is stated in the manuscript, but I feel it could be stressed more, it could be the main “take home” message in the abstract that many biomechanists and anthropologists will find very interesting.

A general comment (cf. Results and Table 1) I have is: why lump the two species of African apes together? Genetically, chimps are closer related to humans than to gorillas. In terms of locomotion chimps and gorillas are closer to one another than to humans, but they are not identical in terms of locomotor behaviour, size and foot anatomy. I suggest treating all species separately. If chimps and gorillas are very similar (and could be lumped) this will then be an additional and interesting finding from this study, but it should not be an assumption. (as a side note, some difference between chimps and gorilla were described, P8 L3-5). If the authors choose to lump the two species, I suggest this should be justified more strongly than is currently the case on P11 L30-31.

P2 L4: mechanical interaction with what?

P2: Axial loading: but leg posture is different and therefore loading may be. Why not load the nonhuman apes in a bent-knee (more dorsiflexed ankle) position?

P2: The conclusion is quite textbook, better focus on a novel finding from this study (see previous comment).

P4: The same maximum load (588 N) was used in all experiments – is equally relevant to all, i.e. were all specimens of a comparable body mass? I assume the chimps would have been lighter than humans, and moreover as quadrupeds their feet would habitually experience lower loads.

P5 L13: “opposite inverting”, do you mean everting?

P6 L5-8: this section (and what leads to it) is one of the crucial findings but a bit abstract when described with words. I believe a schematic drawing or an animation to illustrate this would be useful.

P6 L19-22: why is this VFM a necessity? People walking on pointy stilts cannot generate a VFM by the stance leg, but they can walk. Indeed, it can be counteracted by other moving body parts (e.g., the arms) or just negated in the next step which will create an opposite moment.

P7 L22: the windlass mechanism and basic foot mechanics are being revised and questioned, see e.g., recent papers by Welte and Rainbow on the windlass mechanism (using biplanar X-Ray). I believe it is important to incorporate these recent findings that often go against conventional

knowledge and generally to discuss other studies that have used biplanar X-Ray to study foot function.

P9 L35: n=2 for the bonobos and gorillas, so just give the ages (no means or “range”).

P10 L10-13: How much of the motions can be ascribed to this effect? If the African ape feet were not plantigrade at rest but the human feet were, the potential impact on the results needs to be discussed.

P12 L1: alternative, not alternating

Please do not use kgf but just N as units for force.

The SI is good but quite limited. One suggestion to further increase the value of this study might be to include full kinematics (rotation matrices) and bone models so that the full raw data are available to other scientists who might want to make their own musculoskeletal models? I leave this decision to the author and editors though. I realise than Open Science requires that supporting data are made fully available and I’m not sure if the current spreadsheet file is considered sufficient for the journal.

The figures are good, colours are nice but the figures will also be clear to people with colour blindness.

One limitation is the type of load, esp. its orientation; how realistic is this for the species considered? African apes have ben hips and bent knees so the vertical alignment might be less relevant for them than it is for humans. I do realise there might be reasons to keep alignment constant (for technical as well as reproducibility reasons), but this could be discussed.

Review form: Reviewer 2

Is the manuscript scientifically sound in its present form?

No

Are the interpretations and conclusions justified by the results?

No

Is the language acceptable?

Yes

Do you have any ethical concerns with this paper?

No

Have you any concerns about statistical analyses in this paper?

No

Recommendation?

Reject

Comments to the Author(s)

This study investigated differences in foot bone kinematics of human and other great ape (chimpanzee and gorilla) cadaver feet, in response to static loading. Bi-plane fluoroscopy and

bone surface reconstructions were used to track bone motion as a controlled load was applied to the cadaver feet. The relative motion that occurred under loading was compared between humans and the other great apes. Any differences were then considered in the light of morphological differences, and in relation to adaptations linked to human evolution for bipedal gait. The subject matter is very interesting, and the ability to apply these methods to cadavers from the included species is rare. I believe the experimental work to be well-undertaken, however I have some major concerns with how the data were analysed, and the terminology used.

The use of non-anatomical reference frames for individual bones seems problematic. The authors aligned the reference frame for each bone with the global reference frame, and described bone rotations about the axes of those frames. However, to describe motion of bones or segments it is more normal to create anatomically relevant reference frames that are aligned with the principle axes of the bones themselves. Even though the authors aligned the whole foot with the global coordinate system, individual bones within the foot are not aligned with these axes. Furthermore, the misalignment between individual bones and the global coordinate system is likely to be different in the different species. Therefore the rotations being compared do not represent the same motion in human cadavers vs chimpanzees or gorillas. As a result I do not believe that the comparisons are appropriate.

The terminology used to describe rotations is also not appropriate, in my opinion. First, the authors use inversion/eversion to describe a bone rotating about the anterior-posterior axis of the foot/global coordinate system. Inversion and eversion are specific terms, usually used to describe motion of the forefoot relative to the rear foot. They are generally considered to occur about the subtalar joint axis that is oblique to the longitudinal axis of the foot. Therefore, I do not think that the authors' use of inversion/eversion is correct. Second, the authors describe rotations of bones about the foot's vertical axis as internal/external rotation. Internal/external rotation typically means rotation about the proximo-distal (longitudinal) axis of a bone. It would also be more anatomically correct to calculate rotations using the proximal bone coordinate system as the reference frame (e.g. calcaneus relative to talus). In the authors' current analysis, I don't think the rotations represent the anatomical terms used to describe motion. This is problematic because the main finding of the paper refers to specific anatomical motions of the tibia and calcaneus.

The author's claim that the additional motion observed of the human rear-foot and tibia (referred to by the authors as tibio-calcaneal kinematic coupling) suggests that the human foot-ankle complex is more able to contribute to resisting ground-torques about the vertical axis of the global coordinate system (free moment). However, there is no explanation for why more motion should result in greater torque. In fact it is extremely difficult to mechanistically link the two because we do not know the forces being generated by muscles and ligaments that generate said moment. The link drawn by the authors is far too speculative for me.

The discussion completely ignores the potential for muscles acting around the ankle and on the foot to influence the response of the foot to loading. While it was not possible for the authors to investigate this directly, the discussion should consider how muscles could change the interaction between loading of the foot-ankle and deformation of the foot.

Decision letter (RSOS-210903.R0)

Dear Professor Ogihara

The Editors assigned to your paper RSOS-210903 "Comparative radiographic analysis of three-dimensional innate mobility of the foot bones under axial loading of humans and African great apes" have made a decision based on their reading of the paper and any comments received from reviewers.

Regrettably, in view of the reports received, the manuscript has been rejected in its current form. However, a new manuscript may be submitted which takes into consideration these comments.

We invite you to respond to the comments supplied below and prepare a resubmission of your manuscript. Below the referees' and Editors' comments (where applicable) we provide additional requirements. We provide guidance below to help you prepare your revision.

Please note that resubmitting your manuscript does not guarantee eventual acceptance, and we do not generally allow multiple rounds of revision and resubmission, so we urge you to make every effort to fully address all of the comments at this stage. If deemed necessary by the Editors, your manuscript will be sent back to one or more of the original reviewers for assessment. If the original reviewers are not available, we may invite new reviewers.

Please resubmit your revised manuscript and required files (see below) no later than 30-Jan-2022. Note: the ScholarOne system will 'lock' if resubmission is attempted on or after this deadline. If you do not think you will be able to meet this deadline, please contact the editorial office immediately.

Please note article processing charges apply to papers accepted for publication in Royal Society Open Science (<https://royalsocietypublishing.org/rsos/charges>). Charges will also apply to papers transferred to the journal from other Royal Society Publishing journals, as well as papers submitted as part of our collaboration with the Royal Society of Chemistry (<https://royalsocietypublishing.org/rsos/chemistry>). Fee waivers are available but must be requested when you submit your manuscript (<https://royalsocietypublishing.org/rsos/waivers>).

Thank you for submitting your manuscript to Royal Society Open Science and we look forward to receiving your resubmission. If you have any questions at all, please do not hesitate to get in touch.

on behalf of Dr Jonas Rubenson (Associate Editor) and Kevin Padian (Subject Editor)
openscience@royalsociety.org

Subject Editor Comments to Author:

Thank you for your excellent submission, which both reviewers and the AE appreciated. The writing is clear, the design is straightforward and the illustrations are very well set out. One reviewer is concerned about your "global coordinate system," which I appreciate; they recommend instead movement relative to the two bones of the articulation. I suppose that the identification of a global coordinate system should rest on some kind of actual functional articulation experienced by the living animal, but this is not your study here. I am going to

sustain the AE's recommendation of "reject/resub" but mainly because it will give you time to consider and perhaps calculate the actual articular ranges of the joints, which may not be difficult given the data you already have. Perhaps providing both kinds of data may be of even greater value. However, if you feel that this recommendation is unreasonable, please communicate that in your resubmission. Best wishes.

Associate Editor Comments to Author (Dr Jonas Rubenson):

Dear Dr. Naomichi,

As you will see, the reviewers feel that the study has potential strong interest to the fields of anthropology and anatomy. Nevertheless, Reviewer 2 raises concerns that you will need to address before this work can be considered for publication. Specifically, they challenge the validity of reporting bone motions relative to a global coordinate system as opposed to anatomically relevant reference frames. If the corresponding bones are not aligned the same way in the global coordinate system in each species, then comparing specific motions (e.g. inversion/eversion, plantarflexion-dorsiflexion) is compromised. I agree with the reviewer that assessing the anatomically relevant bone motions is important to this work, especially because it can alter the conclusions regarding comparative foot function. The reviewer also points out that the authors need to revisit their terminology used to describe bone motions as this is not consistent with anatomical standards.

Reviewer comments to Author:

Reviewer: 1

Comments to the Author(s)

This manuscript presents new data on foot bone mobility in two chimpanzees, two gorillas, and four humans, and includes data on five more humans published previously.

This manuscript has straightforward results and is quite descriptive, but the kinematic differences between species have been interpreted well and linked to the overall anatomy.

The methods are well described.

I believe this paper will provide a valuable contribution to the literature.

For me, the most interesting message is that the human foot is actually the more mobile one, which goes against "textbook" knowledge that human feet are stiff and non-human primate feet are flexible. This message is stated in the manuscript, but I feel it could be stressed more, it could be the main "take home" message in the abstract that many biomechanists and anthropologists will find very interesting.

A general comment (cf. Results and Table 1) I have is: why lump the two species of African apes together? Genetically, chimps are closer related to humans than to gorillas. In terms of locomotion chimps and gorillas are closer to one another than to humans, but they are not identical in terms of locomotor behaviour, size and foot anatomy. I suggest treating all species separately. If chimps and gorillas are very similar (and could be lumped) this will then be an additional and interesting finding from this study, but it should not be an assumption. (as a side note, some difference between chimps and gorilla were described, P8 L3-5). If the authors choose to lump the two species, I suggest this should be justified more strongly than is currently the case on P11 L30-31.

P2 L4: mechanical interaction with what?

P2: Axial loading: but leg posture is different and therefore loading may be. Why not load the nonhuman apes in a bent-knee (more dorsiflexed ankle) position?

P2: The conclusion is quite textbook, better focus on a novel finding from this study (see previous comment).

P4: The same maximum load (588 N) was used in all experiments – is equally relevant to all, i.e. were all specimens of a comparable body mass? I assume the chimps would have been lighter than humans, and moreover as quadrupeds their feet would habitually experience lower loads.

P5 L13: “opposite inverting”, do you mean everting?

P6 L5-8: this section (and what leads to it) is one of the crucial findings but a bit abstract when described with words. I believe a schematic drawing or an animation to illustrate this would be useful.

P6 L19-22: why is this VFM a necessity? People walking on pointy stilts cannot generate a VFM by the stance leg, but they can walk. Indeed, it can be counteracted by other moving body parts (e.g., the arms) or just negated in the next step which will create an opposite moment.

P7 L22: the windlass mechanism and basic foot mechanics are being revised and questioned, see e.g., recent papers by Welte and Rainbow on the windlass mechanism (using biplanar X-Ray). I believe it is important to incorporate these recent findings that often go against conventional knowledge and generally to discuss other studies that have used biplanar X-Ray to study foot function.

P9 L35: $n=2$ for the bonobos and gorillas, so just give the ages (no means or “range”).

P10 L10-13: How much of the motions can be ascribed to this effect? If the African ape feet were not plantigrade at rest but the human feet were, the potential impact on the results needs to be discussed.

P12 L1: alternative, not alternating

Please do not use kgf but just N as units for force.

The SI is good but quite limited. One suggestion to further increase the value of this study might be to include full kinematics (rotation matrices) and bone models so that the full raw data are available to other scientists who might want to make their own musculoskeletal models? I leave this decision to the author and editors though. I realise that Open Science requires that supporting data are made fully available and I’m not sure if the current spreadsheet file is considered sufficient for the journal.

The figures are good, colours are nice but the figures will also be clear to people with colour blindness.

One limitation is the type of load, esp. its orientation; how realistic is this for the species considered? African apes have bent hips and bent knees so the vertical alignment might be less relevant for them than it is for humans. I do realise there might be reasons to keep alignment constant (for technical as well as reproducibility reasons), but this could be discussed.

Reviewer: 2

Comments to the Author(s)

This study investigated differences in foot bone kinematics of human and other great ape (chimpanzee and gorilla) cadaver feet, in response to static loading. Bi-plane fluoroscopy and

bone surface reconstructions were used to track bone motion as a controlled load was applied to the cadaver feet. The relative motion that occurred under loading was compared between humans and the other great apes. Any differences were then considered in the light of morphological differences, and in relation to adaptations linked to human evolution for bipedal gait. The subject matter is very interesting, and the ability to apply these methods to cadavers from the included species is rare. I believe the experimental work to be well-undertaken, however I have some major concerns with how the data were analysed, and the terminology used.

The use of non-anatomical reference frames for individual bones seems problematic. The authors aligned the reference frame for each bone with the global reference frame, and described bone rotations about the axes of those frames. However, to describe motion of bones or segments it is more normal to create anatomically relevant reference frames that are aligned with the principle axes of the bones themselves. Even though the authors aligned the whole foot with the global coordinate system, individual bones within the foot are not aligned with these axes. Furthermore, the misalignment between individual bones and the global coordinate system is likely to be different in the different species. Therefore the rotations being compared do not represent the same motion in human cadavers vs chimpanzees or gorillas. As a result I do not believe that the comparisons are appropriate.

The terminology used to describe rotations is also not appropriate, in my opinion. First, the authors use inversion/eversion to describe a bone rotating about the anterior-posterior axis of the foot/global coordinate system. Inversion and eversion are specific terms, usually used to describe motion of the forefoot relative to the rear foot. They are generally considered to occur about the subtalar joint axis that is oblique to the longitudinal axis of the foot. Therefore, I do not think that the authors' use of inversion/eversion is correct. Second, the authors describe rotations of bones about the foot's vertical axis as internal/external rotation. Internal/external rotation typically means rotation about the proximo-distal (longitudinal) axis of a bone. It would also be more anatomically correct to calculate rotations using the proximal bone coordinate system as the reference frame (e.g. calcaneus relative to talus). In the authors' current analysis, I don't think the rotations represent the anatomical terms used to describe motion. This is problematic because the main finding of the paper refers to specific anatomical motions of the tibia and calcaneus.

The author's claim that the additional motion observed of the human rear-foot and tibia (referred to by the authors as tibio-calcaneal kinematic coupling) suggests that the human foot-ankle complex is more able to contribute to resisting ground-torques about the vertical axis of the global coordinate system (free moment). However, there is no explanation for why more motion should result in greater torque. In fact it is extremely difficult to mechanistically link the two because we do not know the forces being generated by muscles and ligaments that generate said moment. The link drawn by the authors is far too speculative for me.

The discussion completely ignores the potential for muscles acting around the ankle and on the foot to influence the response of the foot to loading. While it was not possible for the authors to investigate this directly, the discussion should consider how muscles could change the interaction between loading of the foot-ankle and deformation of the foot.

===PREPARING YOUR MANUSCRIPT===

===PREPARING YOUR REVISION IN SCHOLARONE===

- If you are requesting a discretionary waiver for the article processing charge, the waiver form must be included at this step.
- If you are providing image files for potential cover images, please upload these at this step, and inform the editorial office you have done so. You must hold the copyright to any image provided.
- A copy of your point-by-point response to referees and Editors. This will expedite the preparation of your proof.

- Ensure that your data access statement meets the requirements at <https://royalsociety.org/journals/authors/author-guidelines/#data>. You should ensure that you cite the dataset in your reference list. If you have deposited data etc in the Dryad repository, please include both the 'For publication' link and 'For review' link at this stage.
- If you are requesting an article processing charge waiver, you must select the relevant waiver option (if requesting a discretionary waiver, the form should have been uploaded at Step 3 'File upload' above).
- If you have uploaded ESM files, please ensure you follow the guidance at <https://royalsociety.org/journals/authors/author-guidelines/#supplementary-material> to include a suitable title and informative caption. An example of appropriate titling and captioning may be found at https://figshare.com/articles/Table_S2_from_Is_there_a_trade-off_between_peak_performance_and_performance_breadth_across_temperatures_for_aerobic_scooping_in_teleost_fishes_/3843624.

Author's Response to Decision Letter for (RSOS-210903.R0)

See Appendix A.

RSOS-211344.R0

Review form: Reviewer 1 (Kristiaan D'Août)

Is the manuscript scientifically sound in its present form?

Yes

Are the interpretations and conclusions justified by the results?

Yes

Is the language acceptable?

Yes

Do you have any ethical concerns with this paper?

No

Have you any concerns about statistical analyses in this paper?

No

Recommendation?

Accept with minor revision (please list in comments)

Comments to the Author(s)

I had a number of (mostly minor) comments on the original version of the manuscript. I am happy with how the authors have either dealt with them or rebutted them. They have changed the manuscript accordingly (incl. a new figure and more complete Supplementary Materials).

Regarding my comments #4, #9 and #12, I think it would be good if the authors added a brief statement in the manuscript itself, similar to the rebuttal, for clarity.

I have no further comments.

Review form: Reviewer 2

Is the manuscript scientifically sound in its present form?

Yes

Are the interpretations and conclusions justified by the results?

Yes

Is the language acceptable?

Yes

Do you have any ethical concerns with this paper?

No

Have you any concerns about statistical analyses in this paper?

Yes

Recommendation?

Accept with minor revision (please list in comments)

Comments to the Author(s)

Thank you for responding to my comments regarding the calculations of bone reference frames and bone motion. I believe that inclusion of anatomically defined reference frames improves the data substantially. I do still feel that using terminology such as eversion/inversion and internal/external rotation for individual bone motions is not appropriate and would advise the authors to change this. The author's conventions may be consistent with some prior work, but may be confusing to a wider audience. For example, to describe the calcaneus as in/everting is highly unusual as it is typically the reference frame for this motion.

Decision letter (RSOS-211344.R0)

Dear Professor Ogihara

On behalf of the Editors, we are pleased to inform you that your Manuscript RSOS-211344 "Comparative radiographic analysis of three-dimensional innate mobility of the foot bones under axial loading of humans and African great apes" has been accepted for publication in Royal Society Open Science subject to minor revision in accordance with the referees' reports. Please find the referees' comments along with any feedback from the Editors below my signature.

Please submit your revised manuscript and required files (see below) no later than 7 days from today's (ie 14-Oct-2021) date. Note: the ScholarOne system will 'lock' if submission of the revision is attempted 7 or more days after the deadline. If you do not think you will be able to meet this deadline please contact the editorial office immediately.

on behalf of Dr Jonas Rubenson (Associate Editor) and Kevin Padian (Subject Editor)
openscience@royalsociety.org

Associate Editor Comments to Author (Dr Jonas Rubenson):

Dear Dr. Ogihara and co-authors.

As you will see, the reviewers are largely positive about your revised work. They do provide some relatively minor suggestions that I think will be valuable to address. Regarding R2's comment on the terminology of the bone motions, perhaps it is possible to briefly identify the different conventions used in the literature, and in this way it might alleviate the potential confusion that R2 raises.

Best Regards,
Jonas Rubenson

Reviewer comments to Author:

Reviewer: 1

Comments to the Author(s)

I had a number of (mostly minor) comments on the original version of the manuscript. I am happy with how the authors have either dealt with them or rebutted them. They have changed the manuscript accordingly (incl. a new figure and more complete Supplementary Materials).

Regarding my comments #4, #9 and #12, I think it would be good if the authors added a brief statement in the manuscript itself, similar to the rebuttal, for clarity.

I have no further comments.

Reviewer: 2

Comments to the Author(s)

Thank you for responding to my comments regarding the calculations of bone reference frames and bone motion. I believe that inclusion of anatomically defined reference frames improves the data substantially. I do still feel that using terminology such as eversion/inversion and internal/external rotation for individual bone motions is not appropriate and would advise the authors to change this. The author's conventions may be consistent with some prior work, but may be confusing to a wider audience. For example, to describe the calcaneus as in/everting is highly unusual as it is typically the reference frame for this motion.

===PREPARING YOUR MANUSCRIPT===

===PREPARING YOUR REVISION IN SCHOLARONE===

Author's Response to Decision Letter for (RSOS-211344.R0)

See Appendix B.

Decision letter (RSOS-211344.R1)

Dear Professor Ogihara,

I am pleased to inform you that your manuscript entitled "Comparative radiographic analysis of three-dimensional innate mobility of the foot bones under axial loading of humans and African great apes" is now accepted for publication in Royal Society Open Science.

on behalf of Dr Jonas Rubenson (Associate Editor) and Kevin Padian (Subject Editor)
openscience@royalsociety.org

Appendix A

Dear Reviewer #1,

We greatly appreciate again your through review of our manuscript and valuable and constructive comments. In accordance with your comments and suggestions, we revised the manuscript as follows.

1) For me, the most interesting message is that the human foot is actually the more mobile one, which goes against “textbook” knowledge that human feet are stiff and non-human primate feet are flexible. This message is stated in the manuscript, but I feel it could be stressed more, it could be the main “take home” message in the abstract that many biomechanists and anthropologists will find very interesting.

Answer: We modified the abstract to indicate that the human foot is actually more deformable than in the feet of chimpanzees and gorillas.

2) A general comment (cf. Results and Table 1) I have is: why lump the two species of African apes together? Genetically, chimps are closer related to humans than to gorillas. In terms of locomotion chimps and gorillas are closer to one another than to humans, but they are not identical in terms of locomotor behaviour, size and foot anatomy. I suggest treating all species separately. If chimps and gorillas are very similar (and could be lumped) this will then be an additional and interesting finding from this study, but it should not be an assumption. (as a side note, some difference between chimps and gorilla were described, P8 L3-5). If the authors choose to lump the two species, I suggest this should be justified more strongly than is currently the case on P11 L30-31.

Answer: In the figures, we did not lump the two species of African apes but treated the three species separately. However, to allow statistical comparisons presented in tables, we decided to lump the two species because in terms of locomotion, chimpanzees and gorillas are closer, if not identical, to one another than humans, and the present data also indicated that the foot kinematics due to axial loading is generally more similar between the two species of African apes than between humans and African apes. We tried to better justify this in the revised manuscript.

3) P2 L4: mechanical interaction with what?

Answer: We modified the phrase as follows: mechanical interaction of the human foot with the ground.

4) P2: Axial loading: but leg posture is different and therefore loading may be. Why not load the nonhuman apes in a bent-knee (more dorsiflexed ankle) position?

Answer: If the tibia is tilted in the sagittal plane and the vertical force is applied to the tibia, moment around the ankle joint is generated, which should be counterbalanced by applying additional force to the tibia to maintain static balance of force and moment. This makes the loading condition very different from that of the vertical tibia, making interspecific comparisons impossible. Therefore, the tibia must be set vertical for all three species.

5) P2: The conclusion is quite textbook, better focus on a novel finding from this study (see previous comment).

Answer: We modified the abstract accordingly.

6) P4: The same maximum load (588 N) was used in all experiments – is equally relevant to all, i.e. were all specimens of a comparable body mass? I assume the chimps would have been lighter than humans, and moreover as quadrupeds their feet would habitually experience lower loads.

Answer: Yes, the chimps would have been lighter than humans, but since body weight information was not available for all the specimens including those of humans, the same maximum load was used in all experiments.

7) P5 L13: “opposite inverting”, do you mean everting?

Answer: We deleted the word “opposite”.

8) P6 L5-8: this section (and what leads to it) is one of the crucial findings but a bit abstract when described with words. I believe a schematic drawing or an animation to illustrate this would be useful.

Answer: We added a figure to clarify the explanation.

9) P6 L19-22: why is this VFM a necessity? People walking on pointy stilts cannot generate a VFM by the stance leg, but they can walk. Indeed, it can be counteracted by other moving body parts (e.g., the arms) or just negated in the next step which will create an opposite moment.

Answer: It is true that people on pointy stilts can walk without generating a VFM but is more difficult to counterbalance the yaw moment generated by the swing leg and axial rotation of the trunk. It can be counteracted by arms, but we think generation of a VFM by the innate mobility of the foot unique to humans possibly contributes to facilitate human bipedal locomotion.

10) P7 L22: the windlass mechanism and basic foot mechanics are being revised and questioned, see e.g., recent papers by Welte and Rainbow on the windlass mechanism (using biplanar X-Ray). I believe it is important to incorporate these recent findings that often go against conventional knowledge and generally to discuss other studies that have used biplanar X-Ray to study foot function.

Answer: We added the following sentence in the revised manuscript: It must be noted, however, recent studies suggested that the increase in the stiffness of the foot is more attributable to plantar intrinsic foot muscle activation (Farris et al., 2019, 2020; Kessler et al., 2020; Welte et al., 2021). Further studies are necessary to clarify the detailed mechanism underlying the increased stiffness of the foot enhancing arch recoil at push off in human bipedal locomotion.

11) P9 L35: n=2 for the bonobos and gorillas, so just give the ages (no means or "range").

Answer: Corrected.

12) P10 L10-13: How much of the motions can be ascribed to this effect? If the African ape feet were not plantigrade at rest but the human feet were, the potential impact on the results needs to be discussed.

Answer: The inverted foot is one of the intrinsic morphological differences between the human and African ape feet. Therefore, it is actually impossible to meet completely the same loading condition. Our aim is to clarify the differences in the foot bone motions owing to such differences in the foot morphology.

13) P12 L1: alternative, not alternating

Answer: Corrected.

14) Please do not use kgf but just N as units for force.

Answer: Corrected.

15) The SI is good but quite limited. One suggestion to further increase the value of this study might be to include full kinematics (rotation matrices) and bone models so that the full raw data are available to other scientists who might want to make their own musculoskeletal models? I leave this decision to the author and editors though. I realise that Open Science requires that supporting data are made fully available and I'm not sure if the current spreadsheet file is considered sufficient for the journal.

Answer: As suggested by the reviewer, we now provided full kinematics (rotation matrices) of the bones.

As for the bone models, we have no permissions to make them freely available to public.

16) The figures are good, colours are nice but the figures will also be clear to people with colour blindness.

Answer: We used colors for the graphs in the revised manuscript.

17) One limitation is the type of load, esp. its orientation; how realistic is this for the species considered? African apes have bent hips and bent knees so the vertical alignment might be less relevant for them than it is for humans. I do realise there might be reasons to keep alignment constant (for technical as well as reproducibility reasons), but this could be discussed.

Answer: Okada (2006) presented a picture of a chimpanzee in quiet standing. There, the tibia is quite vertically oriented, and the calcaneus is in contact with the ground. Of course this may not be an average, but the vertical alignment of the tibia may not be less relevant. We explained about this in the revised manuscript.

Dear Reviewer #2,

We greatly appreciate again your through review of our manuscript and valuable and constructive comments. In accordance with your comments and suggestions, we revised the manuscript as follows.

1) The use of non-anatomical reference frames for individual bones seems problematic. The authors aligned the reference frame for each bone with the global reference frame, and described bone rotations about the axes of those frames. However, to describe motion of bones or segments it is more normal to create anatomically relevant reference frames that are aligned with the principle axes of the bones themselves. Even though the authors aligned the whole foot with the global coordinate system, individual bones within the foot are not aligned with these axes. Furthermore, the misalignment between individual bones and the global coordinate system is likely to be different in the different species. Therefore the rotations being compared do not represent the same motion in human cadavers vs chimpanzees or gorillas. As a result I do not believe that the comparisons are appropriate.

Answer: To describe motion of foot bones, it is quite common to define the local coordinate system for each bone such that the axes were aligned with the axes of the global coordinate system both in human (e.g., Okita et al., 2013, Wang et al., 2016) and ape (Holowka et al., 2017) foot studies. As suggested by the reviewer, however, we included the kinematic comparisons based on anatomically relevant local coordinate systems defined based on anatomical landmarks digitized on each bone model, and necessary modifications were made in the revised manuscript. The newly calculated results were generally in consistent with the original results. The paragraph about the calcaneocuboid joint was reorganized since we now have the change in the actual calcaneocuboid joint angle.

2) The terminology used to describe rotations is also not appropriate, in my opinion. First, the authors use inversion/eversion to describe a bone rotating about the anterior-posterior axis of the foot/global coordinate system. Inversion and eversion are specific terms, usually used to describe motion of the forefoot relative to the rear foot. They are generally considered to occur about the subtalar joint axis that is oblique to the longitudinal axis of the foot. Therefore, I do not think that the authors' use of inversion/eversion is correct. Second, the authors describe rotations of bones about the foot's vertical axis as internal/external rotation. Internal/external rotation typically means rotation about the proximo-distal (longitudinal) axis of a bone. It would also be more anatomically correct to calculate rotations using the proximal bone coordinate system as the reference frame (e.g. calcaneus relative to talus). In the authors' current analysis, I don't think the rotations represent the anatomical terms used to describe motion. This is problematic because the main finding of the paper refers to specific anatomical motions of the tibia and calcaneus.

Answer: We used inversion/eversion to describe a bone rotating about the anterior-posterior axis as in other studies (e.g., Okita et al., 2013, Wang et al., 2016). We can certainly change them to supination/pronation, but we had used this terminology in our previous study (Ito et al., 2017). For the sake

of consistency, we prefer to use inversion/eversion. We used internal/external rotation to describe a bone rotating about the vertical axis as in Okita et al.(2013) and Wang et al. (2016). Again, we can change them to adduction/abduction, but we prefer to use internal/external rotation for the same reason. The terms to describe the foot bone movements are actually confusing and differ among researchers. But as long as the terms were clearly specified, they should be acceptable. We would like to ask the reviewer's kind understanding on this matter.

As suggested by the reviewer, we now included the changes in the joint angles due to axial loading, i.e., rotations of the distal bone coordinate system with respect to the proximal bone coordinate system, in the revised manuscript.

3) The author's claim that the additional motion observed of the human rear-foot and tibia (referred to by the authors as tibio-calcaneal kinematic coupling) suggests that the human foot-ankle complex is more able to contribute to resisting ground-torques about the vertical axis of the global coordinate system (free moment). However, there is no explanation for why more motion should result in greater torque. In fact it is extremely difficult to mechanistically link the two because we do not know the forces being generated by muscles and ligaments that generate said moment. The link drawn by the authors is far too speculative for me.

Answer: We are sorry for insufficient explanations. The present study demonstrated that the human tibia was internally rotated if the foot was axially loaded. In addition, our previous study indicated axial torque was generated when the human foot was axially loaded (Seki et al., 2019). When a torque is applied to a deformable structure, it will twist along the axis of the rotation, and the greater the torque is applied, the larger the rotation is generated. Therefore, the deformable foot behaves as a torsional spring when it is axially loaded. We hypothesized that this contributed to generation efficient and stable bipedal locomotion. We explained this in the revised manuscript.

4) The discussion completely ignores the potential for muscles acting around the ankle and on the foot to influence the response of the foot to loading. While it was not possible for the authors to investigate this directly, the discussion should consider how muscles could change the interaction between loading of the foot-ankle and deformation of the foot.

Answer: We are sorry that we failed to discuss about possible contributions of the foot muscles to the interaction between loading and deformation of the foot. We now discussed about this in the revised manuscript.

Appendix B

Dear Reviewer #1,

1) Regarding my comments #4, #9 and #12, I think it would be good if the authors added a brief statement in the manuscript itself, similar to the rebuttal, for clarity.

Answer: We added statements regarding the reviewer's comments #4, #9 and #12 for clarity.

Dear Reviewer #2,

1) Thank you for responding to my comments regarding the calculations of bone reference frames and bone motion. I believe that inclusion of anatomically defined reference frames improves the data substantially. I do still feel that using terminology such as eversion/inversion and internal/external rotation for individual bone motions is not appropriate and would advise the authors to change this. The author's conventions may be consistent with some prior work, but may be confusing to a wider audience. For example, to describe the calcaneus as in/everting is highly unusual as it is typically the reference frame for this motion.

Answer: The terminologies suggested by the reviewer are now specified in parentheses to assist readers.